



**An improved Trajectory-mapped Ozonesonde dataset for the Stratosphere and Troposphere**
**(TOST): update, validation and applications**
Zhou Zang[1], Jane Liu[1], David Tarasick[2], Omid Moeini[2], Jianchun Bian[3], Jinqiang Zhang[3], Anne
M. Thompson[4,5], Roeland Van Malderen[6], Herman G.J. Smit[7], Ryan M. Stauffer[4], Bryan J.
Johnson[8] and Debra E. Kollonige[4,9]
[1]Department of Geography and Planning, University of Toronto, Toronto, Canada
[2]Environment and Climate Change Canada, Toronto, Canada
[3]Key Laboratory of Middle Atmosphere and Global Environment Observation, Institute of
Atmospheric Physics, Chinese Academy of Sciences, Beijing, China
[4]Atmospheric Chemistry and Dynamics Laboratory, NASA Goddard Space Flight Center,
Greenbelt, Maryland, USA
[5]University of Maryland Baltimore County, Baltimore, MD, USA
[6]Royal Meteorological Institute of Belgium, Brussels, Belgium
[7]Institute for Energy and Climate Research: Troposphere (IEK-8), Research Centre Juelich (FZJ),
Juelich, Germany.
[8]NOAA/ESRL Global Monitoring Division, Boulder, Colorado, USA
[9]Science Systems and Applications, Inc., Lanham, MD, USA
*Correspondence: Jane Liu (janejj.liu@utoronto.ca)*
**Abstract**
A global-scale horizontally- and vertically-resolved ozone climatology can provide a detailed
assessment of ozone variability. Here, the Trajectory-mapped Ozonesonde dataset for the
Stratosphere and Troposphere (TOST) ozone climatology is improved and updated to the recent
decade (1970s-2010s) on a grid of $5° \times 5° \times 1$ km (latitude, longitude, and altitude) from the surface
to 26 km altitude, with the most recent ozonesonde data re-evaluated following the ASOPOS-2
guidelines (GAW Report No. 268, 2021). Comparison between independent ozonesonde and
trajectory-derived ozone shows good agreement in each decade, altitude, and station, with relative



differences (RD) of 2-4% in the troposphere and 0.5% in the stratosphere. Comparisons of TOST
with aircraft and two satellite datasets, the Satellite Aerosol and Gas Experiment (SAGE) and the
Microwave Limb Sounder (MLS), show comparable overall agreement. The updated TOST
outperforms the previous version with higher data coverage in all latitude bands and altitudes and
14-17% lower RD compared to independent ozonesondes, employing twice as many ozonesonde
profiles and an updated trajectory simulation model. Higher uncertainties in TOST are where data
are sparse, i.e., over the southern high latitudes and the tropics, and before the 1980s, and where
variability is high, i.e., at the surface and upper troposphere and lower stratosphere (UTLS).
Caution should therefore be taken when using TOST in these spaces and times. TOST captures
global ozone distributions and temporal variations, showing an overall insignificant change of
stratospheric ozone after 1998. TOST offers users a long record, global coverage, and high vertical
resolution.
**1. Introduction**
The global ozone distribution and its long-term changes at different altitudes, longitudes, and
latitudes are critical to understanding global ozone variability and its interactions with climate
change. While the ozone trends themselves can indicate the impact of changes in climatic
dynamics (Hassler et al., 2008), or chemistry, including the effect of the Montreal Protocol
(Steinbrecht et al., 2017), long-term horizontally- and vertically-resolved ozone are needed for
prescribing, evaluating and refining ozone simulations in climate models (Hassler et al., 2018),
and to quantify changes in radiative forcing and projecting reliable future climate scenarios
(Nowack et al., 2015).

Balloon-borne ozonesondes are the principal source of trend-quality long-term records of

ozone profiles below ~18 km (Tarasick et al., 2021). However, the horizontal and temporal



coverages of ozonesondes are limited by the sparse distribution of the stations (less than 100
worldwide) and their low launch frequency (1-3 times/week) (Liu et al., 2013a). The In-Service
Aircraft for a Global Observing System (IAGOS) program has measured ozone profiles worldwide
since 1994 via the instruments onboard a number of commercial aircraft, with high sampling
frequency at some airports (Thouret et al., 1998). However, sampling is unevenly distributed both
spatially and temporally because the flights are constrained by commercial airlines' operation
schedules. Satellite observations have the advantage of providing large-scale 3-dimensional ozone
data with consistent quality. However, satellite data are provided for the stratosphere only or for
troposphere with limited vertical resolution (6–10 km) (Worden et al., 2007; Liu et al., 2010;
Tarasick et al., 2019b) caused by uncertainties for satellites to retrieve tropospheric ozone through
the large stratospheric ozone burden (Bhartia, 2002). A number of studies have developed long-
term (since the 1980s) ozone climatologies by combining ozone data from ozonesondes and/or
multiple satellite instruments (McPeters et al., 2007; McPeters and Labow, 2012; Hassler et al.,
2018; Bodeker et al., 2021; Bognar et al., 2022), but these are generally zonally-averaged.
Chemistry–climate models are also used to develop 3-dimensional ozone data fields, especially
for long-term, global-scale simulations (Eyring et al., 2010; Chen et al., 2018); these models
present our best understanding of processes controlling ozone variationsbut still suffer from large
uncertainties regarding the inventories, parameterizations, radiation transport schemes, and
simulation of the atmospheric circulations and systems (Young et al., 2018; Wild et al., 2020;
Griffiths et al., 2021; Zeng et al., 2022).

Liu et al. (2013a, b) constructed a long-term 3-dimensional global-scale ozone dataset using

a trajectory-mapping method, extending sparse ozonesonde measurements and filling gaps in the
spatial domain by backward and forward trajectory simulations. The trajectory-mapping method
assumes the ozone mixing ratio in the same air parcel along each trajectory path is constant for



several days, which is reasonable given that the lifetime of ozone in most of the troposphere and
stratosphere ranges from weeks to months (Jacob, 1999). The result is a global dataset that is
independent of satellite measurements and photochemical modeling processes. The trajectory-
mapping method can be characterized as a meteorologically-guided interpolation method, which
necessarily carries more information than conventional statistical interpolation methods (Stohl et
al., 2001). In addition, the trajectory-derived ozone data cover higher latitudes (to 90ºN and 90ºS)
and a longer time period (since the 1960s) (Liu et al., 2013b). The Trajectory-mapped Ozonesonde
dataset for the Stratosphere and Troposphere (TOST) Version 1 is available from 1965-2012 at the
World Ozone and UV Data Centre (WOUDC, https://woudc.org/archive/products/ozone/vertical-
ozone-profile/ozonesonde/1.0/tost/, last access: Jan 29, 2024), and has been successfully applied
in model evaluation (Skeie et al., 2020; Badia et al., 2021), ozone and climate trend studies
(Polvani et al., 2017; Gaudel et al., 2018; Gulev et al., 2021), as a background ozone climatology
(Xu et al., 2018; Moeini et al., 2020, and for tropospheric ozone burden estimation (Griffiths et al.,

2021).

There have been several important developments since the publication of the first version of

TOST data in 2013 (Liu et al. 2013a, b), which we refer to as TOST Version 1, or TOST-v1. An
improved version of TOST, namely TOST-v2, is necessary for the following reasons. Firstly, there
are some 50,000 new ozone profiles, many from newly established ozonesonde stations (see
Section 2.1). These new ozonesonde data permit updating TOST to 2021, providing 3-dimensional
ozone information through the 2010s. Secondly, data from many ozonesonde stations have been
updated to higher-quality versions. An important source of uncertainty in TOST-v1 is possible
biases in station records due to instrument changes and/or changes in operating procedures.
Homogenized time series are now available from the Harmonization and Evaluation of Ground
Based Instruments for Free Tropospheric Ozone Measurements (HEGIFTOM) project for over 40



ozonesonde stations (Table S1). For these records, biases due to instrument changes, sensing
solution, and preparation changes have been corrected, to reduce the overall uncertainty from 10-
20% to 5-10% (Smit and Thompson, 2021). This effort to improve data quality also uncovered an
apparent change of bias at stations flying one type of sonde (Stauffer et al., 2020; 2022); 14 global
ozonesonde stations (the bolded stations in Table S1) have shown an apparent drop-off of 2-4 %
in stratospheric ozone and total ozone column since circa 2013, due to a possible instrument artifact.
This is the subject of ongoing research (e.g. https://gml.noaa.gov/annualconference/abstracts/78-
230424-A.pdf, last access: Jan 29, 2024). For these stations, ozone measurements above 40 hPa
(~20 km) are not recommended for trend calculations.  We need therefore to exclude data above
40 hPa for the affected profiles in constructing TOST. Thirdly, the version 4.9 of the Hybrid Single-
Particle Lagrangian Integrated Trajectory (HYSPLIT) model (Draxler and Hess, 1998) used for
trajectory simulation has been improved and updated to version 5.2. Here, we address the
mentioned issues and construct an improved and updated TOST using the most state-of-the-art
HYSPLIT and most updated ozonesonde data. While Liu et al. (2013a, b) validated TOST-v1 with
ozonesonde data at 20 selected stations, TOST-v2 is validated against the ozonesonde data at all
141 stations individually with the trajectory-mapped approach omitting the input from the station
being tested. In addition, comparisons are made with the IAGOS measurements in the troposphere,
and with two limb-viewing satellite sensors, the Satellite Aerosol and Gas Experiment (SAGE)
and the Microwave Limb Sounder (MLS), in the stratosphere. This more comprehensive validation
and associated uncertainty analysis demonstrates the improved quality of TOST-v2, and also
provides some caveats for users of TOST.

123         In the following, Section 2 explains the data sources and the improved trajectory-mapping

methodology. Section 3 presents independent validations, comparisons with satellite data, and
improvements compared to TOST-v1, as well as the uncertainties in TOST-v2. Based on TOST-



v2, we characterize global ozone variations in the troposphere and stratosphere, and show stagnant
stratospheric ozone variation since the late 1990s in Section 4, followed by conclusions in Section

128 5.


**2. Data and Methods**
**2.1 Ozonesonde data**
Ozonesonde data over 1970-2021 at 141 ozonesonde stations worldwide (Figure 1) were
downloaded from the World Ozone and Ultraviolet Radiation Data Centre (WOUDC,
https://woudc.org/archive/Archive-NewFormat/OzoneSonde_1.0_1/), or where available,
homogenized data from Southern Hemisphere ADditional OZonesondes (SHADOZ,
https://doi.org/10.57721/SHADOZ-V06, last access: Jan 29, 2024) and HEGIFTOM
(https://hegiftom.meteo.be/datasets/ozonesondes, last access: Jan 29, 2024). The homogenized
ozonesonde stations from HEGIFTOM include ozonesonde stations from the SHADOZ network
(Thompson et al., 2017; Witte et al., 2017; 2018), the Canadian network (Tarasick et al., 2016),
the US network (Sterling et al., 2018), the Network for the Detection of Atmospheric Composition
Change (NDACC) and several individual stations (Van Malderen et al., 2016; Witte et al., 2019;
Ancellet et al., 2022), with an overall accuracy of 3-5% in both the stratosphere and troposphere.
Ozonesonde data from the Beijing Nanjiao Meteorological Observatory (116.47°E, 39.81°N) in
Beijing, China, are provided by the Institute of Atmospheric Physics (IAP), Chinese Academy of
Sciences. The ozone profiles at Beijing are measured by the Brewer-Mast type GPSO3 ozonesonde
and the IAP electrochemical concentration cell (ECC) ozonesonde, which are in fair agreement
with commercial ECC ozonesondes (Wang et al., 2003; Xuan et al., 2004; Bian et al., 2007) in
both laboratory and field experiments (Zhang et al., 2021; Zeng et al., 2023). In total, data from
43 more stations were used in this version of TOST than in TOST-v1 (Liu et al., 2013b).





Figure 1a provides an overview of the distribution of the ozonesonde stations, the number of
profiles, and the beginning year for every station. Most of the stations with data before the 1980s
are located in North America, Europe, and East Asia. The majority of the stations in the Southern
Hemisphere start measurement in the 1990s or later, and so the Southern Hemisphere contains a
smaller number of ozone profiles than in the Northern Hemisphere. Figure 1b shows that the total
number of ozonesonde profiles per year has almost doubled since the 1990s and reached a
maximum in the late 2000s with over 3000 profiles per year. Since then, the available amount of
ozonesonde profiles has declined slightly to 2000-3000 profiles per year. The average annual
number of profiles per station slightly increased since the 1990s and has stabilized at about 40
profiles per year.
All the ozonesonde profiles were processed into 1-km vertical resolution by integrating and
averaging the ozone volume mixing ratio in 1-km layers from the ground level. The ozonesonde
data above 26 km were excluded as the data above this height show large uncertainties at mid- and
high-latitudes (Fioletov et al., 2006).

**2.2 Trajectory simulation**
Forward and backward trajectories in four days were calculated every 6 hours using the
version 5.2 HYSPLIT model (Stein et al., 2015). HYSPLIT was driven by the reanalysis of hourly
meteorological data from the National Centers for Environmental Prediction/National Center for
Atmospheric Research (NCEP/NCAR), which has a horizontal resolution of 2.5° by 2.5° in latitude
and longitude and 17 vertical levels from the surface to 10 hPa (Kalney et al., 1996). The length
of the trajectories influences the spatial coverage and accuracy of the ozone mapping. Generally,
uncertainties increase rapidly along the trajectories, with typical errors of about 100–200 km day$^{-1}$
(Stohl, 1998). Trajectories have horizontal uncertainties of 350–400 km after 3 days and 600-1000



km after 4 days in the Northern Hemisphere (Engström and Magnusson, 2009). Trajectories show
typical vertical deviations of about 200, 800, and 1000 m after 2, 4, and 6 days in the stratosphere,
and even greater uncertainties in the troposphere (Stohl and Seibert, 1998). Therefore, to limit
trajectory errors, 4-day trajectories were used herein, following previous studies (Tarasick et al.,
2010; Liu et al., 2013 a, b).

**2.3 Three-dimensional ozone mapping based on ozonesonde profiles and trajectories**
Ozone mixing ratios from each sounding at the 26 levels were assigned to the corresponding
forward and backward trajectory paths. These ozone values at positions every 6 hours along the 4-
day backward and forward trajectories (32 positions for each level) were averaged in bins of 5°
latitude and 5° longitude, for each 1-km altitude for every month. This bin size corresponds both
to the typical uncertainties of 4-day trajectories discussed above, and to the typical ozone
correlation length (500-1500 km) in the troposphere and the stratosphere (Liu et al., 2009).
Ozonesonde profile data and trajectories in both the troposphere and stratosphere were used to
represent the exchanges of ozone between the troposphere and stratosphere in ozone climatology.
Based on this mapping, TOST was generated at 26 altitude levels in monthly means for each
decade from the 1970s to the 2010s and in annual means for each year from 1970 to 2021.

Errors in the mapped data can come from trajectory errors, and from ignoring ozone chemistry

(production and loss) along the transport pathway and deposition in the surface layer (Liu et al.,
2013a). Differences between the results of backward and forward trajectory mapping can provide
a measure of these errors, since in the absence of such errors the results of forward-only and
backward-only trajectory mapping should be identical. Therefore, mappings from the forward-
only and backward-only trajectories were compared as an initial quality check. Figure S1 shows
monthly means (January and July) in 2000 at 3-4 km and 19-20 km, for forward-only and



backward-only mapping. In general, the differences between the two mappings are commonly less
than 15% and have no distinct pattern, indicating that trajectory errors, and those from ozone
chemistry and deposition, are not systematic. These modest differences between forward-only and
backward-only trajectory-mapped ozone fields also validate the reliability of this trajectory-
mapping method; both backward and forward trajectories, therefore, were combined in TOST to
achieve better averages and higher spatial coverage.
The resulting ozone fields are given in two altitude coordinates (altitude above sea level and
altitude above ground level) for users' convenience. In addition, three ozone climatology datasets
are generated based on trajectories from ozonesonde observations in both the troposphere and
stratosphere, trajectories from observations only in the troposphere (troposphere-only) and
trajectories from observations only in the stratosphere (stratosphere-only). Examples presented in
this paper all use ozone mapping based on trajectories from observations in both the troposphere
and stratosphere, with altitudes above sea level. For this coordinate system, both ozonesonde
profiles and mapped data necessarily begin at the altitude of the surface, leaving the levels below
as null.

**2.4 Validations of TOST**
To comprehensively validate TOST, several validations and comparisons were conducted.
**2.4.1 Ozonesonde profiles for validation**
The first method is to compare the actual ozone profile at each of the ozonesonde stations with
the trajectory-derived ozone profile for that station without the input of that station itself. This
method is computationally intensive, as the trajectory mapping must be re-calculated (with data
for all stations except one), for each ozonesonde station, but it directly tests the reliability of
deriving ozone concentrations at a location by integrating the contributions via trajectories from





surrounding sites, which is the essential assumption of the trajectory-mapping method. We refer
to this set of data that selectively excludes the local data at each station as "Traj-derived".
**2.4.2 Satellite ozone profile data**
TOST is further compared with two well-known satellite limb sounder datasets, the Satellite
Aerosol and Gas Experiment (SAGE) and the Microwave Limb Sounder (MLS).
SAGE II was launched into a 57-degree inclination orbit on board Earth Radiation Budget Satellite
(ERBS), and was in operation from 1984–2005. Using the highly accurate solar occultation
technique, SAGE can resolve layers in the middle and upper troposphere at 1-km vertical
resolution (Kent et al., 1993), with the highest accuracy over the 20-45 altitude range (Cunnold et
al., 1996). Here we use the Version 7.0 SAGE II ozone mixing ratio
(https://sage.nasa.gov/missions/about-sage-ii/, last access: Jan 29, 2024) in the 1980s and 1990s
for the comparison.
The MLS, onboard the Aura satellite, can measure stratospheric ozone profiles with a vertical
resolution of about 3 km. MLS observes microwave radiances that are both emitted and absorbed
by the atmosphere. The retrieval is more complex, and uses the optimal estimation approach. Here
we use the Version 5.0 MLS ozone mixing ratio
(https://disc.gsfc.nasa.gov/datasets/ML2O3_005/summary?keywords=ML2O3_005, last access:
Jan 29, 2024) in the 2000s and 2010s for the comparison.
**2.4.3 Aircraft ozone profile data**
The IAGOS network (https://www.iagos.org/, last access: Jan 29, 2024) has been measuring ozone
profiles worldwide since 1994 via dual-beam ultraviolet absorption monitors onboard commercial
aircraft (Petzold et al., 2015), with an accuracy of about ± (2 nmol mol$^{-1}$ + 2%) (Nédélec et al.,
2016). Ozone monitors are calibrated annually to a reference analyser at the Bureau Internationale
des Poids et Mesures (BIPM), and also compared every 2 hours to an in-flight ozone calibration

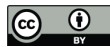



source. Generally good agreement is found between IAGOS profiles and ozonesondes, with
positive biases for the sondes of 5-10% (Tilmes et al., 2012; Zbinden et al., 2013; Staufer et al.,
2013, 2014; Tanimoto et al., 2015; Tarasick et al., 2019b), making IAGOS ozone suitable for the
validation of TOST. Here, the IAGOS ozone profiles were processed into 1 km layers from sea
level and matched with the TOST ozone for each level to examine the performance of TOST in
the troposphere.

**3. Validations and comparisons of TOST**

**3.1 Validations with ozonesonde observations**

First, we show the overall comparison in monthly mean ozone profile between ozonesonde and
trajectory-derived values without the inputs of the stations being tested (Traj-Derived), from all
the existing stations at selected altitude levels. Note that the full TOST dataset would be better
than "Traj-Derived ozone", especially at the sampling locations because the input of the local
station is included in the full TOST data. The three altitude levels are selected to present the overall
accuracy of TOST in the lower troposphere (ozone concentration at 0-50 ppbv,), the upper
troposphere (ozone concentration at 50-150 ppbv) and the stratosphere (ozone concentration
at >150 ppbv).
Figure 2a-e shows the overall tropospheric ozone comparisons between independent
ozonesonde (Sonde-Observed) and Traj-Derived ozone in the entire study period (Figure 2a-c) and
each decade (Figures 2d). Overall, the Sonde-Observed and Traj-Derived ozone concentrations
agree well in the lower troposphere (Figure 2a), with a correlation coefficient (R) of 0.69 and a
root mean square (RMS) difference [square root of the mean of squared individual differences] of
7.5 ppbv, a low bias (0.7 ppbv) and RD (1.8%) [where RD is the relative difference 100 × (TOST
ozone - ozonesonde ozone)/ozonesonde ozone)]. The linear fit for the entire study period shows a



slope of 0.99. In the upper troposphere (Figure 2b), the agreement between the Sonde-Observed
and Traj-Derived ozone concentration is moderately lower, with a linear fitting coefficient of 1.01
and RMS of 21.1 ppbv, and higher bias (2.9 ppbv) and RD (4.0%) than those in the lower
troposphere. This lower agreement in the upper troposphere owes to greater influence of
stratosphere-to-troposphere (STE) in the upper troposphere, where trajectories by the Lagrangian
dispersion model (such as HYSPLIT) show substantially increased deviations due to the strong
turbulence and convection (Stohl et al., 2002). The positive bias may imply that STE is slightly
overestimated in HYSPLIT, as the comparison between the Sonde-Observed and troposphere-only
Traj-Derived ozone concentrations shows a clear underestimation (with RD of -9% to -5%) in the
upper troposphere (Figure S2). In the stratosphere (Figure 2c), the overall agreement between the
Sonde-Observed and Traj-Derived ozone concentrations has a linear fitting coefficient of 0.97 and
an RMS of 416.9 ppbv. The small bias is of higher magnitude (11.1 ppbv) to that in the troposphere
but this is much smaller relative to stratospheric ozone concentrations; the RD is only 0.5%,
indicating higher reliability of Traj-Derived in the stratosphere.
This validation method compares ozonesonde station data with Traj-Derived ozone, i.e., the
ozone found by averaging trajectories that come from other stations, some of which will have
higher ozone, and some lower. The average difference results from an imbalance in the distribution
of meteorological trajectories, and this is confirmed by detailed analysis. For example, before the
1990s, fitting coefficients were smaller than 1 and Rs were smaller than 0.60 (Figure 2d) in the
lower troposphere, indicating a tendency to underestimate the Traj-Derived ozone in the lower
troposphere. After the 1990s, owing to the additional ozonesonde measurements provided by
SHADOZ in the tropics, the underestimation of Traj-Derived ozone in the lower troposphere is
greatly reduced and the linear fitting coefficient is very close to 1 (and Rs increased to > 0.71).
Similarly, with the additional ozonesonde measurements after the 1990s, the Rs in the upper



troposphere increased from < 0.50 to > 0.58. In all decades, the agreement between Sonde-
Observed and Traj-Derived ozone in the stratosphere is the best, with Rs of ~0.97 and linear
coefficients of 0.99. The RD in each decade is small (-0.3% - 1.4%), indicating no systematic
underestimation or overestimation in the stratospheric Traj-Derived data. However, in the upper
troposphere, Traj-Derived ozone tends to be overestimated, with RD of 0.6-4.5%.
Figure 3 examines how the RD between the ozonesondes and Traj-Derived ozone values
varies with altitude, presenting the frequency distributions of RD across all stations, at every other
altitude level and in each decade. The distributions of RD show little skewness in every other
altitude and decade, indicating no systematic bias during the study period. The overall interquartile
ranges (25-75%) of RD are between -30 and 30%, with the lowest interquartile ranges of RD (-10
to 10%) in the stratosphere and middle and lower troposphere. Higher interquartile ranges of RD
appear in the 13-19 km altitude range, where the upper troposphere-lower stratosphere (UTLS)
region is located, and are due to the large vertical gradients of ozone concentrations in the UTLS
and the variability of the tropopause (Millan et al., 2023). The surface (boundary layer) ozone,
however, shows a positive bias of the median, in all decades, of up to 12%, suggesting that TOST,
which neglects ozone chemistry and deposition, often overestimates ozone concentration there.
Figure 4 exemplifies comparisons in vertical profiles between Sonde-Observed and Traj-
Derived ozone profiles at individual stations in different seasons. Four stations with sufficient data
coverage (>15 years) were selected from the Antarctic coastal region (Syowa), Europe
(Hohenpeissenberg), North America (Boulder), and East Asia (Beijing). The decadal mean (1990s
and 2000s) profiles in January and July are used to compare the performance of Traj-Derived
ozone profiles in boreal winter and summer. In general, the Traj-Derived profiles can capture the
vertical ozone variation in different seasons, with good correlation (R > 0.99) and high accuracy
(bias < 100 ppbv, RD < 10%) in comparison to the independent ozonesonde profiles. The Syowa



comparison shows a larger bias, but much of this is due to the fact that in the 1990s this station
launched the Japanese KC-79 carbon-iodine sonde, while other stations in the Southern
Hemisphere launched ECC sondes; the Traj-Derived profiles would therefore be expected to be
10-20% higher in the troposphere and about 5% higher in the lower stratosphere (Smit and Kley,
1998). The excellent agreement in tropospheric ozone at Hohenpeissenberg is likely due to
frequent and dense European ozonesonde observations; similar cases also are seen at Uccle,
Payerne, and Praha. Larger discrepancies are shown near the planetary boundary layer (PBL) and
UTLS, as the simulated trajectories over these regions have more uncertainties (Stohl and Seibert,
1998; Sicard et al., 2019), and ozone chemistry and deposition are potentially important in the PBL
at time scales similar to that of the longer trajectories (four days).

**3.2 Comparisons with satellite data**
To compare with satellite data, we first validated the Traj-Derived ozone profiles against
ozonesonde measurements. The corresponding validation was conducted for the satellite data of
SAGE and MLS in the same period and location. The sets of ozonesonde, Traj-Derived and
satellite data were selected only when all three datasets were available in the same month, decade,
and gridpoint, so to ensure that both the Traj-Derived and satellite data could be independently
evaluated by the ozonesondes. Figures 5a-d show the vertical RD of the Traj-Derived and SAGE
ozone. Compared to SAGE, Traj-Derived ozone concentrations agree with the ozonesondes better
in the troposphere (<12 km), with the RD generally < 20%. Above 11 km, Traj-Derived and SAGE
ozone concentrations have comparable RD of 10-25% between 12-20 km, and less than 5% above
20 km. In the 12-20 km range, SAGE ozone agrees better with the ozonesondes, particularly in the
1980s.

Figures 5e-h compare the vertical RD of the Traj-Derived and MLS ozone values. The MLS



profiles are validated only above the altitude recommended (261 hPa, Livesey et al., 2022). In the
lowermost stratosphere, from 12-17 km, MLS shows comparable or better performance than Traj-
Derived ozone, while above 17 km the RD of MLS ozone is higher by 0.62-11.88% than that of
Traj-Derived ozone, particularly in boreal summer (JJA).

It is of course expected that TOST would outperform satellite instruments in measurements

below the tropopause, as satellite measurements are hampered by the large stratospheric ozone
burden that satellite instruments must look through, but these comparisons suggest that even above
15 km, where SAGE and MLS are considered most reliable (Wang et al., 2002; Kremser et al.,
2020; Livesey et al., 2022), TOST can provide comparable or better accuracy.

Figures 6 show time series of the vertical variation of monthly RD from 16-26 km between

Traj-Derived and SAGE ozone from 1985-2005, and between Traj-Derived and MLS ozone from
2005-2019. SAGE ozone data are reliable above 20 km (Kremser et al., 2020), having a mean RD
of about -10-10%, similar to that of Traj-Derived ozone. SAGE ozone concentrations are lower
than the Traj-Derived ozone by 5 to 10% between 16 and 20 km (Figure 6f), as Traj-Derived ozone
overestimates the ozonesondes by 9 to 15% (Figure 6e) while SAGE ozone underestimates the
ozonesondes by -7 to -1% (Figure 6d). Over the MLS period from 2005 to 2019, TOST ozone at
all altitudes between 16 and 26 km agrees with independent ozonesondes better than during the
SAGE period (Figures 6h and 6k vs. Figures 6b and 6e). Accordingly, the Traj-Derived ozone
concentrations show good agreement with MLS ozone above 22 km, but are lower than MLS
ozone below 20 km (Figures 6i and 6l), as MLS generally overestimates ozone concentrations
below 20 km (Figures 6g and 6j).

Figure S4 compares the RMSE of Traj-Derived and satellite ozone in different latitude zones

from 16-26 km. Compared to SAGE in the 1990s, the Traj-Derived ozone has comparable RMSEs
in the Northern Hemisphere, yet higher RMSEs in the Southern Hemisphere, due to the fewer





ozonesonde stations there. MLS ozone also shows lower RMSEs in the Southern Hemisphere, but
higher RMSEs in the Northern Hemisphere.
Figure S5 compares monthly average ozone mixing ratios of Traj-Derived ozone with
corresponding SAGE and MLS averages, above 16 km, in two seasons. The monthly average
values correlate very well, with R = 0.94-0.98, for both instruments and both seasons. Seasonally,
Traj-Derived ozone is slightly higher than either SAGE or MLS ozone in DJF (linear fitting
slope >1; RD between 1 and 3%), but markedly lower than MLS in JJA (linear fitting slope 0.91-
0.92; RD -8 to -9%).
Table S2 summarizes the evaluation of both Traj-Derived and satellite ozone against the
ozonesondes over 16-26 km. The Traj-Derived and SAGE ozone values show high correlation (R
= 0.95 or greater in all cases), and the Traj-Derived comparison shows RDs of -1% to +2% in the
1980s and 1990s, but only -0.3% to +0.4% in the 2000s and 2010s. By contrast, the SAGE
comparison shows RDs of -4% to +0.5%, while the MLS comparison shows RDs of -2% to +11%.

**380 3.3 Comparisons with aircraft observations**

We also compare TOST ozone with the IAGOS dataset, in the lower troposphere at 0-50 ppbv,
from 1994-2021 (Figure 7). Note that this comparison is between the full TOST (not Traj-derived)
and IAGOS datasets here. TOST ozone values are generally higher than IAGOS with a mean bias
of 2.2 ppbv and R of 0.49, but RDs (5.8%) and RMS (8.8 ppbv) are low. The linear fit has a slope
of 1.03. The two ozone datasets employ different measurement techniques and atmospheric
sampling (Petetin et al., 2018). Previous studies have reported that IAGOS ozone values are
systematically lower than ozonesonde values, typically by 5-10% in the free troposphere (Tilmes
et al., 2012; Zbinden et al., 2013; Staufer et al., 2013, 2014; Tanimoto et al., 2015; Tarasick et al.,
2019b). The comparisons in Figure 6 are consistent with these earlier estimates, as the RD (Figure





3d) indicates that IAGOS measurements average 6% lower than TOST, with only slight variation
(5-8%) when the comparison is made by decade. In the upper troposphere at 6-10 km, however,
the IAGOS measurements are on average 12% lower than TOST, with also slight variation (11-
12%) between decades (Figure S3).

**3.4 Improvements in the new version**
The improvements in TOST-v2 are attributed to the increased amount and improved quality of
ozonesonde data, as well as the improved trajectory simulation and ozone mapping. Because more
ozonesonde stations and more ozonesonde data have become available since the 1990s or 2000s
(Table S1), more ozone profiles were used in constructing TOST-2, leading to improved data
density. Table S3 summarizes the data coverage, the number of ozonesonde stations and
ozonesonde profiles used for TOST-v2 and TOST-v1. The data coverage is defined as the ratio of
the number of gridpoints with valid annual means to the total number of gridpoints in the
corresponding latitudinal zone. The number of ozonesonde stations, compared to Liu et al. (2013b),
increases in all latitudes by ~50%, and the total number of ozonesonde profiles used is doubled.
Data coverage increases as well, in all latitude bands, by 5-15% (Table S3) and in all altitudes by
a maximum of 10% (Figure S6).
In addition to the data density, the data quality was also improved in TOST-v2. Figure 8a-b
shows the distributions of ozone concentrations in TOST-v2 and TOST-v1 at the lowest level (0-1
km) for the 2000s. Over the Antarctic, gaps are observed only in the new TOST data. This is more
reasonable for the sea-level data because the altitude over the Antarctic is over 1 km (Figure S7a),
where ozone trajectories should not appear at 0-1 km. Therefore, the spatial distributions of ozone
are clearly improved with this topography correction in TOST-v2 compared to TOST-v1, which
could    be    attributed    to    the    updated    terrain    file    since    HYSPLIT    v5.0



(https://www.arl.noaa.gov/hysplit/hysplit-model-updates/). Over the eastern Pacific, marked with
an ellipse in Figure 8a, b, TOST-v1 shows higher ozone concentrations than TOST-v2 by 30%
(Figure 8c). Compared to the ozonesonde measurement at 0-1 km in the 2000s in these two regions
(Davis station for the Antarctic and Easter Island station for the eastern Pacific), TOST-v2 agrees
better with ozonesondes than TOST-v1, indicating better representation of ozone distributions
(Figure S7b).
With reference to spatial distributions at 19-20 km in the 2000s, Figure 8d-e shows that in the
Antarctic and the tropical eastern Pacific, TOST-v1 values show higher concentrations than TOST-
v2 (Figure 8f). Figure S7c compares ozone concentrations from ozonesonde, TOST-v2, and TOST-
v1 at 19-20 km in the 2000s at an Antarctic station (Syowa) and a tropical station (Bogota).
Compared to TOST-v1, TOST-v2 ozone values show a better agreement with the ozonesonde
measurement. The difference between TOST-v2 ozone and ozonesonde measurements is 10% and
29% in Syowa and Bogota stations, while in TOST-v1, ozone concentrations at these stations show
24% and 39% differences (Figure S7c).
In summary, TOST has been improved in TOST-v2 with higher spatial coverage, improved
description of ozone spatial distributions, and a better agreement with ozonesonde measurements
in both the troposphere and stratosphere.

**3.5 Uncertainty analysis**
As noted in Section 2.3, the ozone concentrations in each of the TOST gridpoints (or bins) in a
month are determined by the ozone concentrations along all the trajectories passing through that
gridpoint in that month. Therefore, an estimate of the random uncertainty of TOST may be
obtained from the standard error of the mean in each bin. Note that this may not be a true estimate
of the standard error, as some bins may contain more than one value from an individual trajectory,





depending on wind speed, and so these values are not independent and our standard error
calculation is biased low.

For convenience, given the large range of ozone concentrations between the stratosphere and

troposphere, we use the ratio of the standard error to the mean in that bin, SE/Mean, expressed
in %. The standard error is proportional to the variability of the ozone values in a bin (i.e. the
standard deviation) and inversely proportional to the square root of the number of data values.
Thus in general, the more trajectories passing a gridpoint, the more data points for that gridpoint
and the lower the standard error for that gridpoint. Figure 9 shows the SE/Mean and the number
of samples in January and July of the 2000s at 3-4 km and 19-20 km. Generally, the Southern
Hemisphere shows higher SE/Mean values (> 10%) than the Northern Hemisphere (< 6%), which
reflects the large number (>100) of ozone soundings in the Northern Hemisphere, especially over
North America and Europe. However, near the equator, despite the higher sampling rate, the
SE/Mean still is as high as 15%. Compared to the stratospheric level (19-20 km), the tropospheric
level (3-4 km) shows an overall higher SE/Mean. SE/Mean varies less with season in the
stratosphere than in the troposphere. For example, at 3-4 km, the SE/Mean in January is generally
<7% but becomes >10% in July in the Northern Hemisphere, and vice versa in the Southern
Hemisphere. This is likely due to more vertical motion in the PBL (Stohl and Seibert, 1998; Sicard
et al., 2019) so that ozone in some bins comes from multiple altitude levels, as well as increased
photochemistry and biomass burning. Stratospheric intrusions to the lower troposphere are more
frequent in boreal spring and summer than in winter (Terao et al., 2008; Greenslade et al., 2017),
and can be responsible for much of the variability at 3-4 km (Tarasick et al., 2019a).

To quantify the uncertainties of TOST ozone in different altitudinal and latitudinal zones, and

in different seasons and decades, we calculated the Normalized Root Mean Squared Error
(NRMSE) of the monthly ozone mixing ratio between ozonesonde and Traj-Derived ozone over





1970-2021 (Figure 10). Among altitudes, the highest NRMSE values appear at 9-10 km and over
the tropopause region, and the second highest NRMSE at the surface, while the lowest NRMSE
values are in the lower to middle troposphere (3-6 km) and stratosphere (19-26 km), consistent
with Figure 4. There is considerable variation in NRMSE with latitude; the NRMSEs in the
southern high latitudes (90-60S) and the northern tropics (0-30N) are higher than in other
latitudinal zones. This could reflect higher horizontal gradients of ozone (e.g. stations in or outside
the ozone hole) in the southern high latitudes or biases between ECC sondes and other types (the
Indian and Japanese sondes) in the northern tropics. By season, the NRMSE varies slightly with a
lower value in March-April-May than in other seasons. After the 1990s, the NRMSEs are reduced
markedly compared to the 1980s and 1970s, likely related to the improved data coverage in the
later period. This overview provides caveats regarding where (surface and UTLS, the northern
high latitudes and tropics) and when (before the 1990s) more caution is advised when using TOST.

**4. Global ozone spatial-temporal variations observed from TOST**
**4.1 Ozone spatial variations in the troposphere and stratosphere**
As a 3-dimensional ozone dataset, TOST can depict both horizontal and vertical ozone
distributions, as well as long-term ozone timeseries. Figure 11 shows distributions of decadal mean
TOST ozone at 3-4 km and 19-20 km in four seasons of the 2000s. At 3-4 km in the troposphere,
ozone concentrations are higher over the continent in the Northern Hemisphere, especially in
MAM and DJF (>50 ppbv), reflecting the ozone production from the photochemical reactions of
anthropogenic and natural emissions. In addition, the continental outflow from the southern US
(in MAM) and the biomass burning-produced ozone in southern Africa (in JJA and SON) are well
captured and in agreement with satellite observations (Fishman et al., 1990; Ebojie et al., 2016).
At 19-20 km in the stratosphere (Figure 11e-h), ozone concentrations are higher near the poles



than in the tropics, due to the impact of the Brewer–Dobson circulation. The North Pole has higher
ozone concentrations than the South Pole in DJF and MAM, and vice versa in JJA and SON,
reflecting the seasonality of the Brewer-Dobson circulation. Also at 19-20 km, the ozone
concentrations are lower over Asia in JJA (Figure 11f) than in other seasons, reflecting the
transport of ozone by Asian summer monsoon from the tropics (Gettelman et al., 2004; Bian et al.,

2020).

Although trajectory mapping fills in much of the spatial domain, large gaps can still be found,

particularly in the tropics, where ozone soundings are less dense. Since some applications require
a default ozone value at all gridpoints, a smoothed ozone dataset is also provided for the decadal
mean ozone in each month and the annual mean ozone, by fitting the maps at each level to a linear
combination of spherical functions (Liu et al., 2013b). As shown in Figures 11i-p, small-scale
variations and extreme values are reduced in the smoothed ozone fields, while broad patterns of
the ozone distribution are retained, making these smoothed maps valuable for qualitative
visualization of the spatial, seasonal, and decadal variations in ozone at different altitudes. They
should, however, be used for any kind of quantitative analysis with great caution, as these highly
interpolated data, where gaps exist in the unsmoothed TOST dataset, are necessarily far from any
original measurement and the degree to which they represent the true ozone value is doubtful. For
example, erroneous conclusions have been inferred from the smoothed TOST-v1 output over the
tropics, with very limited observations before 1998, where the smoothed data were mostly
interpolated from higher latitudes (Chipperfield et al., 2022). In addition, smoothing, as noted,
removes small-scale variations and extreme values, and does so whether they are real or not. The
smoothed dataset has not been quantitatively evaluated in any way.

Figure 12a-d shows the latitude-altitude distribution of TOST ozone in each season averaged

over 1970-2021. The steep changes in ozone concentration from <100 to >500 ppbv in the vicinity



of the tropopause (the black lines in Figure 12a-d, calculated from the NCEP/NCAR reanalysis)
are well captured. Due to the Brewer-Dobson circulation, ozone concentrations above the
tropopause increase with latitude from the tropics to the poles, which is also well reflected in the
latitude-altitude distribution. TOST ozone concentrations are higher in spring (600-800 ppbv) than
in the other seasons (< 500 ppbv) over northern midlatitudes (45-60°N) at about 12-13 km, which
reflects the stronger Brewer-Dobson circulation in spring (Holton et al., 1995). Figure 12e shows
the monthly mean TOST ozone time series from 1970 to 2021, averaged over 30-70°N at each
level. Clear seasonal cycles are well captured every year.

### 4.2 Long-term trend in stratospheric ozone

One of the advantages of TOST is its long-term coverage, which enables investigation of variations
in ozone back to the 1970s. One application is to study stratospheric ozone changes, as it is
important to assess stratospheric ozone recovery (or lack thereof) since the implementation of the
Montreal Protocol and its amendments (Fang et al., 2019). While this is commonly done with
individual ozonesonde time series, it is challenging to assess how well individual long-term station
changes represent regional or global variations. Combining data records from sparse and widely
separated ozonesonde sites involves implicit assumptions about their representativeness. With
meteorological trajectory mapping, each original ozonesonde measurement is assigned a trajectory
which describes its representativeness, and the TOST averages are therefore weighted according
to the representativeness of each measurement. While this is subject to trajectory errors and the
fact that coverage is incomplete (Table S3), unless trajectory errors are non-random, it should
produce a better result than simple averaging of sonde station data by geographic region.

Figure 13 shows the area-weighted annual averages of ozone concentrations at 21-22 km and

24-25 km from 1970 to 2021; averages were taken over all gridpoints from 30°-70°N with





available data throughout all years (~ 70% of gridpoints). The 3-year running means are also shown
with the time series. The ozone time series at both levels captures the ozone depletion in the early
1990s from the effects of the 1991 eruption of Mt. Pinatubo (McCormick et al., 1995; Tang et al.,
2013; Dhomse, et al., 2015) and the recovery in the latter part of the 1990s. In addition, these
updated TOST time series show that stratospheric ozone since 2000 changed little, despite the
decline in stratospheric chlorine since then. From Figure 14, there is an insignificant trend in the
ozone concentrations at 21-22 km (by 0.6 ppbv/year) and 24-25 km (by -1.9 ppbv/year) from 1998
to 2021, indicating little change of stratospheric ozone, despite the fact that 25 years have passed
since peak stratospheric chlorine. Using long-term satellite data, Bognar et al. (2022) also find
stratospheric ozone largely unchanged in the last two decades, which, they suggest, is related to
asymmetries and long-term variability in the Brewer-Dobson circulation. Such observations of the
variation in stratospheric ozone are essential to verifying the expected stratospheric ozone recovery
under the Montreal Protocol.

**5. Conclusions**
An improved TOST dataset has been generated from 1970 to 2021 based on the updated
ozonesonde profiles at 141 ozonesonde stations from WOUDC, SHADOZ, HEGIFTOM and
NDACC. The updated TOST was derived by combining the 4-day forward and backward
trajectories from each ozonesonde profile, which were driven by the most state-of-the-art
HYSPLIT model (v5.2) and NCEP reanalysis data (NNRP-1). The monthly mean ozone, decadal
mean ozone in each month (January to December from the 1970s-2010s) and annual mean ozone
(1970-2021) are provided in 3-dimensional grids of 5º × 5º × 1 km (latitude, longitude, and
altitude). Spatially-smoothed maps are also provided by decadal mean in each month and annual
mean for qualitative visualization, model initialization, and other applications with caution. For



user convenience, the TOST data in coordinates from the sea level and from the surface level are
both generated, and separate ozone climatology datasets are generated based on trajectories from
ozonesonde in both the troposphere and stratosphere, trajectories from ozonesonde only in the
troposphere and trajectories from ozonesonde only in the stratosphere. Statistics (standard error,
number of samples) are also provided.

Comprehensive validation of TOST-v2 was conducted. At all the ozonesonde stations used,

trajectory-derived ozone profiles without the input of the station itself were compared with the
corresponding ozonesonde profiles at the stations. The overall comparison between the
ozonesonde and trajectory-derived ozone shows good agreement in both the troposphere (R = 0.56-
0.69, RD = 2-4%) and stratosphere (R = 0.97, RD = 0.5%) in each decade and in all decades' mean
(Figure 2). The frequency distribution of RD at different altitudes shows interquartile ranges of
RD between -30 to 30%, with the lowest interquartile ranges of RD (-10 to 10%) in the stratosphere
and lower troposphere, and no systematic bias except in the surface layer (Figure 3). The patterns
of ozone profiles at individual stations are also well captured and quantified, with R > 0.76 and
RD of 2-8% (Figure 4). Larger discrepancies are shown near the PBL and UTLS, especially for
coastal stations where the trajectory-derived ozone may be biased by trajectories from the
continent (Tarasick et al., 2010).

The comparison between TOST and satellite data, i.e., SAGE in the 1980s and 1990s, and

MLS in the 2000s and 2010s, illustrates that TOST data have comparable accuracy with the
satellite data in the stratosphere, while in the troposphere TOST is markedly superior (Figure 5).
In different latitude zones and decades, TOST performs comparably with SAGE and MLS data as
well (Figure 6). TOST-v2 was also directly compared to MOZAIC-IAGOS ozone profiles over the
period 1994-2021 from the surface to 5 km. Despite the systematic difference between MOZAIC-
IAGOS and ozonesonde measurements, the two ozone datasets agree well in each decade and in



all decades' mean for lower troposphere (RD =5-8%, Figure 7) and upper troposphere (RD =11-
12%, Figure S3).

Compared to the previous version of TOST (TOST-v1, Liu et al. 2013a and b), this new TOST,

TOST-v2, is improved in two major aspects. Firstly, the record is extended to 2021 and data
coverage is increased by as much as 15%, as more ozone profiles and 43 additional ozonesonde
stations are used in constructing the new version of TOST. Secondly, the spatial distribution of
ozone has better agreement with ozonesonde measurements in both the troposphere and
stratosphere over regions of Antarctica and the eastern Pacific, with RD decreased by > 50%.
The uncertainties of TOST are largely dependent on the availability of ozonesonde data. Higher
uncertainties are found before the 1990s, as global coverage is sparse in the tropics before
SHADOZ. Higher uncertainties also appear at southern high latitudes and in the northern tropics,
with NRMSE there being 35-78% higher than at northern midlatitudes (Figure 10), likely because
of greater ozone variability there, although biases between ozonesonde types may also contribute.
TOST data at the PBL and UTLS have higher standard error and twice the NRMSE compared to
other altitude levels; the former is due to more small-scale processes in the PBL while the latter is
related to the large ozone gradient and the dynamic variation of the tropopause.

TOST can capture global ozone distributions in the troposphere and stratosphere (Figures 11

and 12), showing horizontal and vertical variations, the continental outflow, and the gradient of
ozone concentration near the tropopause. TOST can also reflect the seasonal variations in ozone
concentrations near the vicinity of the tropopause. The time series of the updated TOST shows the
stagnant recovery but overall insignificant change of stratospheric ozone after 1998 (Figure 13),
which agrees well with studies using satellite-based and model-based ozone datasets (Bognar et
al., 2022).

It is anticipated that this updated and improved TOST dataset can benefit future studies, owing





to its long record, global coverage, and high vertical resolution. We expect that it will be a useful
dataset for trend studies, especially in the free troposphere, and also in the stratosphere, given the
excellent long-term stability of the global ozonesonde network (Stauffer et al., 2022). We caution,
however, that users should keep in mind the assumptions and limitations of the data product as
described here.

**Author contribution**
J. L. and D.T. conceptualized and designed this study. Z.Z. performed data process, analysis, and
composed the first draft. All the coauthors contributed substantially to this study in making
ozonesonde measurements, processing, calibrating, and archiving the ozonesonde data, and
providing constructive and valuable suggestions to and comments on the manuscript. All the co-
authors approved the submission of this paper.

**Code and data availability**
The ozonesonde data used in this study can be obtained from the WOUDC
(https://woudc.org/archive/Archive-NewFormat/OzoneSonde_1.0_1/), SHADOZ
(https://doi.org/10.57721/SHADOZ-V06) and HEGIFTOM (https://hegiftom.meteo.be/datasets
/ozonesondes). The trajectory model HYSPLIT (Version 5.2) is from the NOAA Air Resources
Laboratory (http://www.arl.noaa.gov/ready.html), driven by the NCEP/NCAR reanalysis data
from the NOAA/OAR/ESRL PSD, Boulder, Colorado, USA, at https://www.ready.noaa.gov/data/
archives/reanalysis/. The aircraft data can be accessed from IAGOS network
(https://www.iagos.org/). The two satellite data for comparison, the SAGE II (Version 7.0) and
the MLS (Version 5.0), are obtained from https://sage.nasa.gov/missions/about-sage-ii/ and
https://disc.gsfc.nasa.gov/datasets/ML2O3_005/summary?keywords=ML2O3_005, respectively.



We are in the process of making the TOST available at the WOUDC website. TOST data

currently are available on request from the authors.

**Competing interests**

The authors declare that they have no conflict of interest.

**Special issue statement**

This article is part of the special issue "Tropospheric Ozone Assessment Report Phase II (TOAR-II) Community Special Issue (ACP/AMT/BG/GMD inter-journal SI)".

**Acknowledgements**

We thank many for their dedication to WOUDC, SHADOZ, and HEGIFTOM, making ozonesonde data accessible. We also thank SAGE II and MLS team for their ozone data for comparison. We acknowledge the HYSPLIT team for the trajectory model. Z. Z. and J. L. acknowledge the financial support from Natural Science and Engineering Council of Canada (Grant No. RGPIN-2020-05163); J. B. and J. Z. from the National Natural Science Foundation of China (Grant No. 42293321).

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

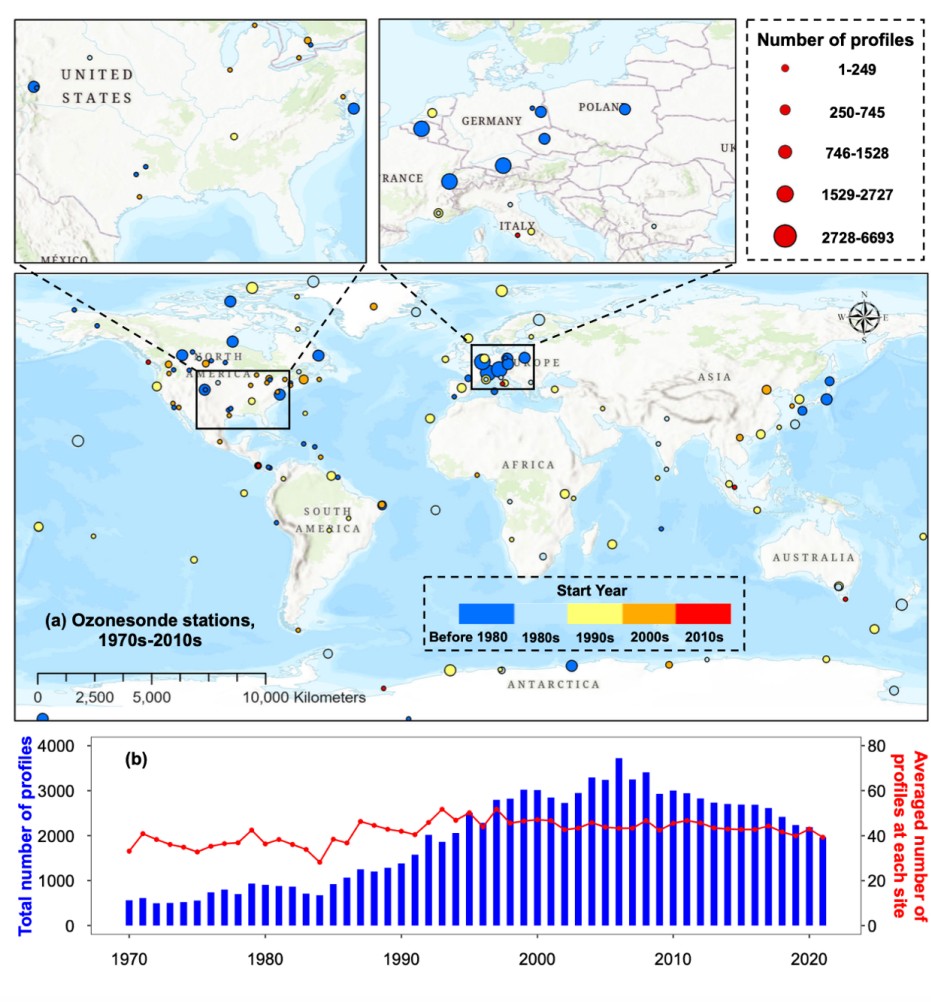



Figure 1. (a) Global distribution of ozonesonde stations used in this study to construct TOST-v2.
Station details are provided in Table S1. The size and color of the dots indicate the total number
of sounding profiles and the start year of the measurement time series. (b) The total number of
profiles per year (left y-axis, blue bars) and the average number of profiles per site and per year





(right y-axis, red dots and line) from 1970 to 2021.

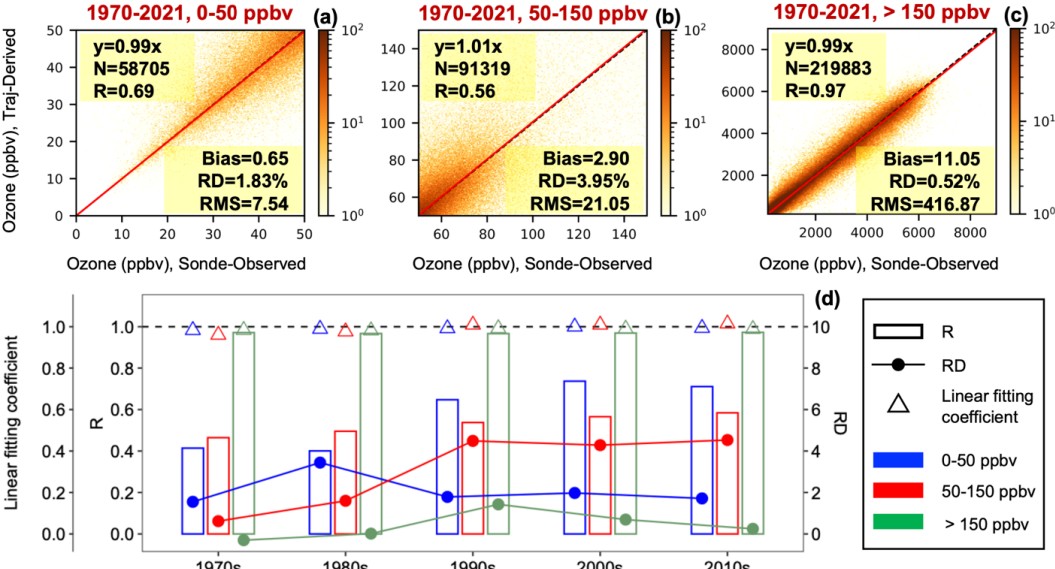


Figure 2. (a-c) Comparison of monthly average tropospheric ozone mixing ratios from
ozonesondes (Sonde-Observed) and trajectory-derived TOST data (Traj-Derived) for the entire
study period of ozone concentration at 0-50 ppbv, 50-150 ppbv and >150 ppbv. Solid red lines
represent the linear fitting line (with the intercept set to 0) and dashed black lines denote the 1:1
axis. N is the total number of data points, R is the correlation coefficient, Bias is the overall average
difference in monthly mean values [Traj-Derived ozone - Sonde-Observed ozone, in ppbv], RD is
the relative difference in % [100 × (Traj-Derived ozone - Sonde-Observed ozone)/ Sonde-
Observed ozone)], and RMS is the root mean square difference in ppbv). Note that Traj-Derived
ozone at each station is derived without input from the station itself; that is, Traj-Derived represents
an ensemble of 141 separate computations of TOST, each one withholding a single validation
station. (d) the R (bars), RD (dots and lines) and linear fitting coefficient (with the intercept set to
0; triangles) between the Traj-Derived ozone and Sonde-Observed ozone by decade. The dashed
line denotes where the linear fitting coefficient is 1.





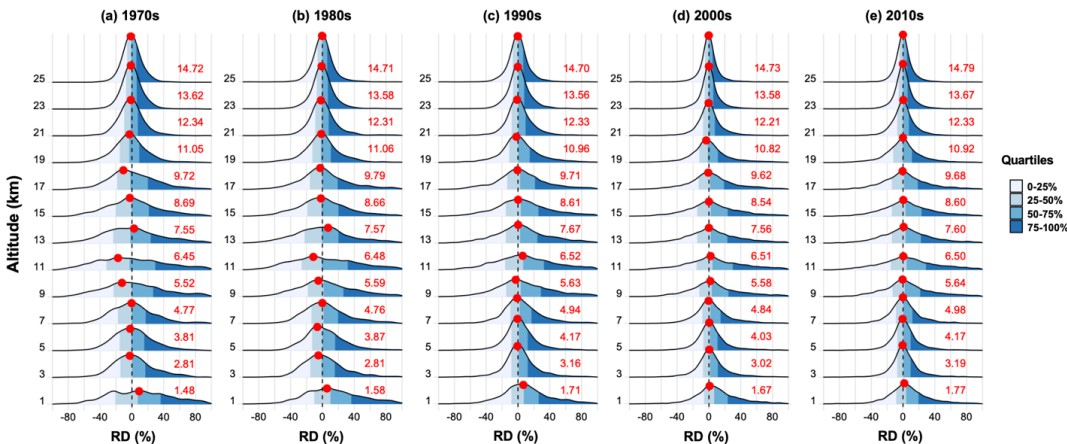

Figure 3. The relative difference (RD) of the monthly ozone mixing ratios between ozonesonde and Traj-Derived data by altitude in the 1970s, 1980s, 1990s, 2000s and 2010s, respectively. The frequency distribution of RD at every other altitudes is shown (y-axis: frequency in %, x-axis: RD in %), with the colors denoting the 4 quartiles of RD. The dashed line indicates zero difference in RD. The red dot and number represent the maximum frequency and the corresponding frequency value in %, respectively.

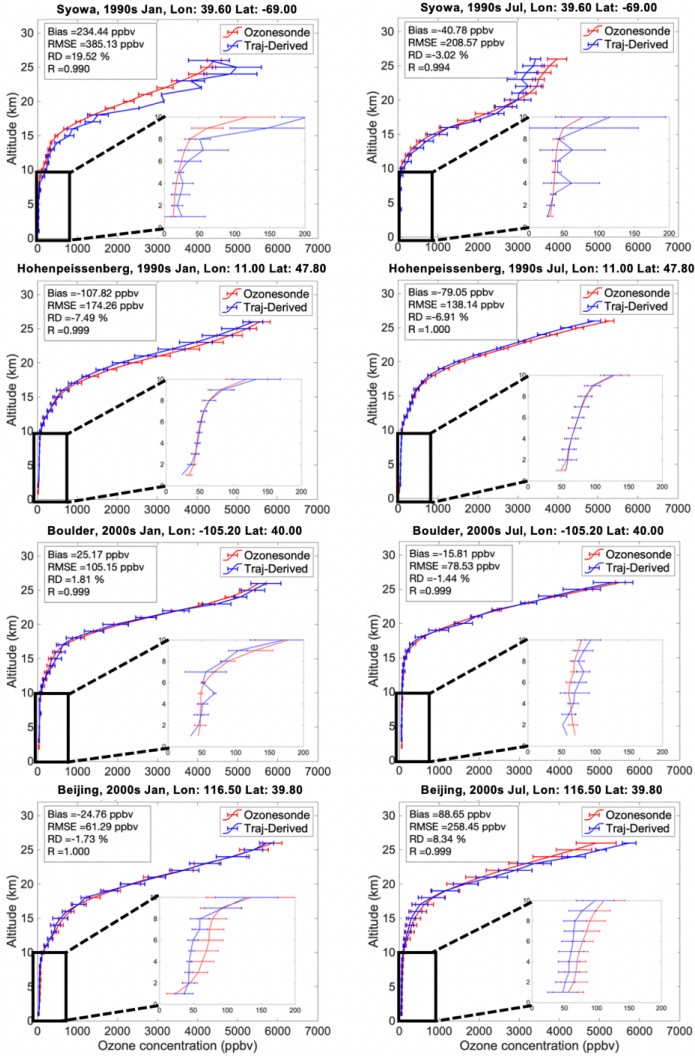

Figure 4. Decadal monthly mean ozone profiles at Syowa and Hohenpeissenberg in January and July 1990s, and at Boulder and Beijing in January and July 2000s. The red line denotes ozonesonde ozone and the blue line denotes trajectory-derived ozone without the input from the station itself. The error bar is ±2 times the standard error of the mean (equivalent to 95 % confidence limits on the averages). To better compare the difference of ozone profiles in the troposphere, a zoom-in window from 0-10 km is provided in each sub-figure.

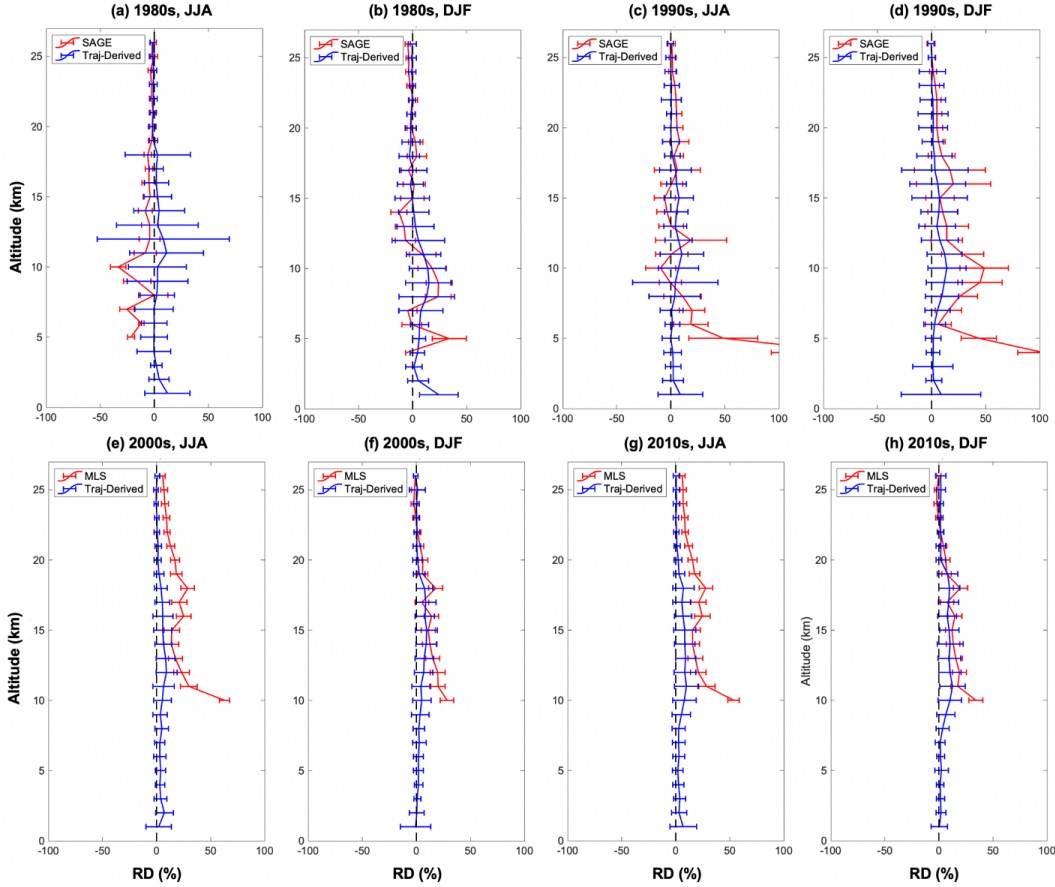

Figure 5. (a-d) Mean relative difference (RD) of the monthly ozone mixing ratios between the Traj-Derived and ozonesonde data (blue line) and between the SAGE and ozonesonde data (red line) in JJA (June-July-August) and DJF (December-January-February), in the 1980s and 1990s. (e-h) Decadal seasonal mean RD between the trajectory-derived and ozonesonde data (blue line) and between the MLS and ozonesonde data (red line) in JJA and DJF, in the 2000s and 2010s. The error bars represent ±1 standard deviation of the seasonal mean RDs at each altitude in each decade.



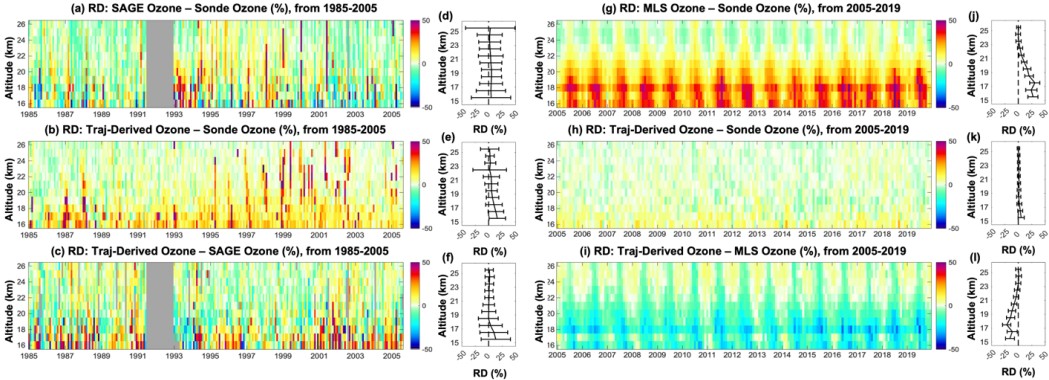

Figure 6. (a). The relative difference (RD) between ozonesonde and SAGE ozone data in each month and at each altitude during 1985-2005 over 16-26 km [RD = 100×(SAGE ozone – ozonesonde ozone)/ozonesonde ozone, in %]. The mean RD over 1985-2005 at each level is shown on the right, where the error bars represent the standard deviation of the monthly RD over 1985-2005. Note that the Pinatubo-affected SAGE profiles are excluded during July 1991- December 1992 (filled with gray color). (b) same as (a), but for the RD between ozonesondes and Traj-derived data, [RD = 100×(Traj-derived ozone – ozonesonde ozone)/ozonesonde ozone, in %]. (c) same as (a), but for the RD between Traj-derived and SAGE ozone data [RD = 100 × (Traj-derived ozone – SAGE ozone)/(0.5 × Traj-derived ozone + 0.5 × SAGE ozone), in %]. (d-f) the averaged RD by altitude corresponding to (a-c). (g-l) same as (a-f), but for the period of 2005-2019 and the satellite measurements are from MLS ozone.



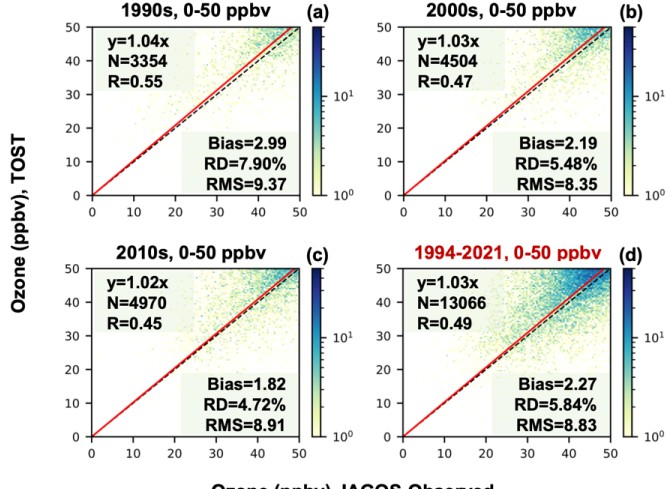

992

Figure 7. The comparison of monthly ozone mixing ratios between IAGOS-observed (x-axis

labeled: IAGOS-Observed) and TOST data (y-axis labelled: TOST) by decade (a-c) and for the

entire study period (d) of ozone concentration at 0-50 ppbv. Solid red lines represent the linear

fitting line (with the intercept set to 0) and dashed black lines denote the 1:1 axis. N is the total

number of data points, R is the correlation coefficient (unitless), Bias is the difference in monthly

mean values [TOST ozone - IAGOS ozone, unit: ppbv], RD is the relative difference [$100 \times$ (TOST

ozone - IAGOS ozone)/($0.5 \times$ TOST ozone + $0.5 \times$ IAGOS ozone)], and RMS the root mean square

difference (unit: ppbv).



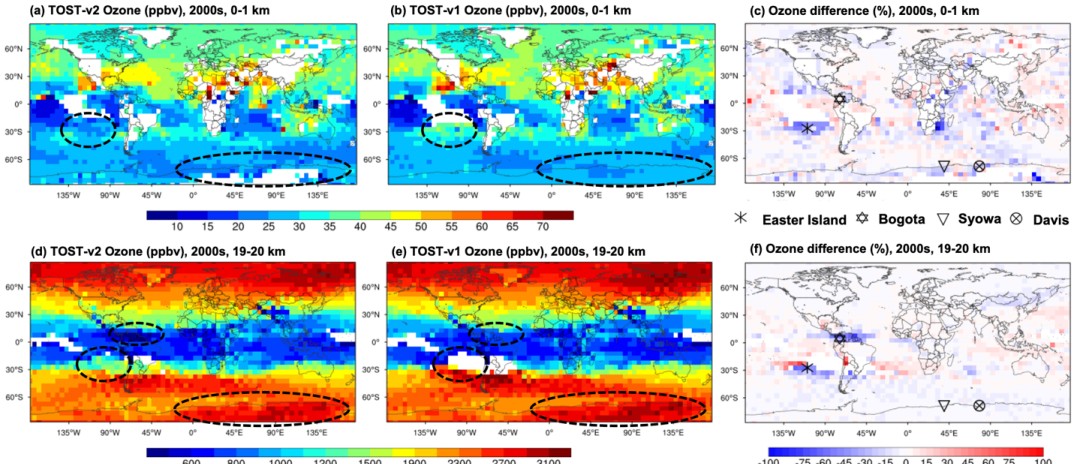

Figure 8. (a, b) The global distributions of ozone in TOST-v2 (a) and TOST-v1 (b) over 0-1 km in the 2000s. (d, e) The same as for (a, b), but over 19-20 km. The dashed circles indicate regions with large differences between the two versions. (c) The global distributions of RD between TOST-v2 and TOST-v1 [RD = 100 × (TOST-v2 – TOST-v1)/(0.5 × TOST-v2 + 0.5 × TOST-v1), in %] over 0-1 km in the 2000s. (f) The same as for (c), but over 19-20 km. The markers indicate the positions of Davis, Easter Island, Bogota and Syowa stations.



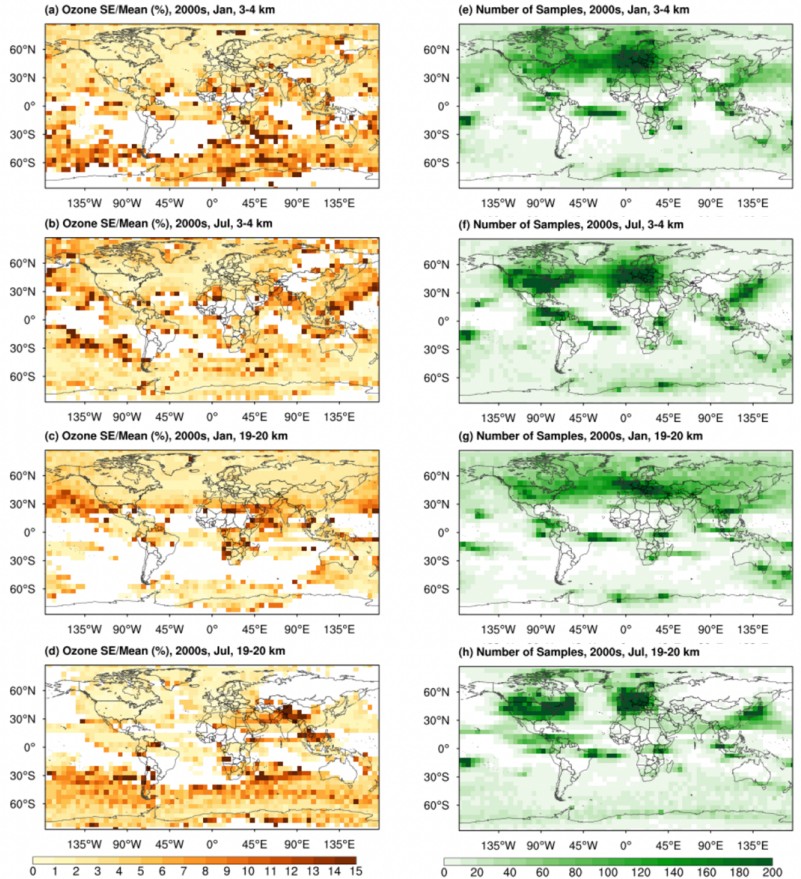



Figure 9. (a-d) Global distribution of the standard error of the mean (left panels, in %) for the
decadal monthly mean ozone in January and July 2000s at 3-4 km (a and b) and 19-20 km (c and
d). (e-h) the same as (a-d), but for the number of samples in each 5 × 5° bin.



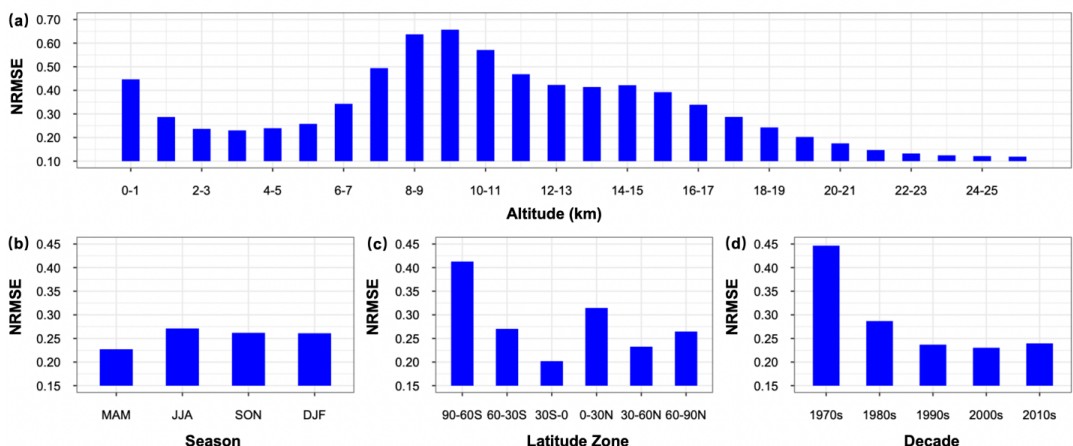

Figure 10. The Normalized Root Mean Squared Error (NRMSE, unitless) of TOST over 1990-2021 by altitude, and the average NRMSE over all altitudes by (a), season (b), latitudinal zone (c), and decade (d). The NRMSE is calculated as the RMS difference of monthly ozone mixing ratio between ozonesondes and Traj-Derived ozone divided by the mean ozone mixing ratio from ozonesondes measurements.




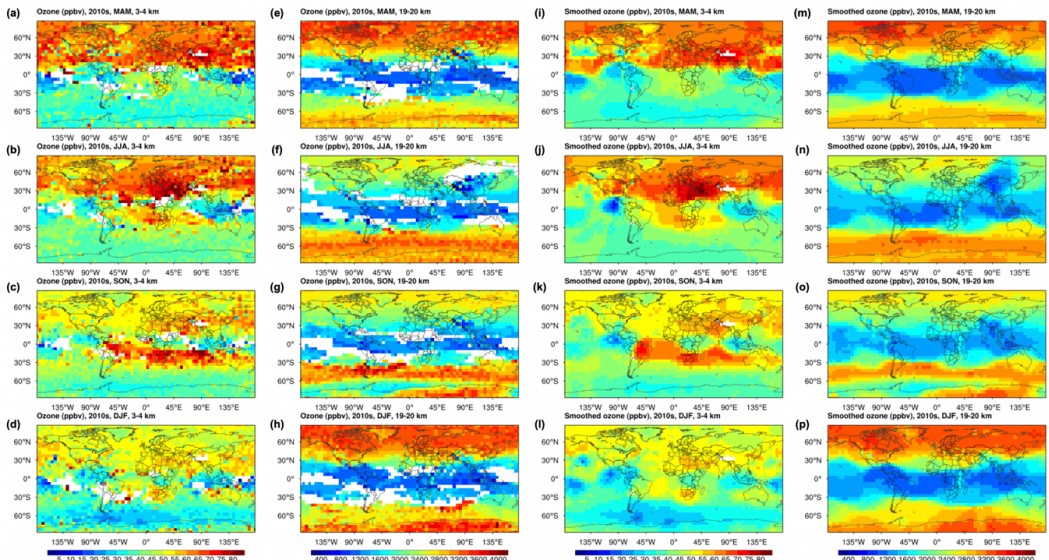


Figure 11. Global distribution of decadal mean TOST ozone at 3-4 km and 19-20 km in MAM
(March-April-May), JJA (June-July-August), SON (September-October-November) and DJF
(December-January-February) in the 2010s (a-h), and the corresponding smoothed TOST ozone
(i-p).




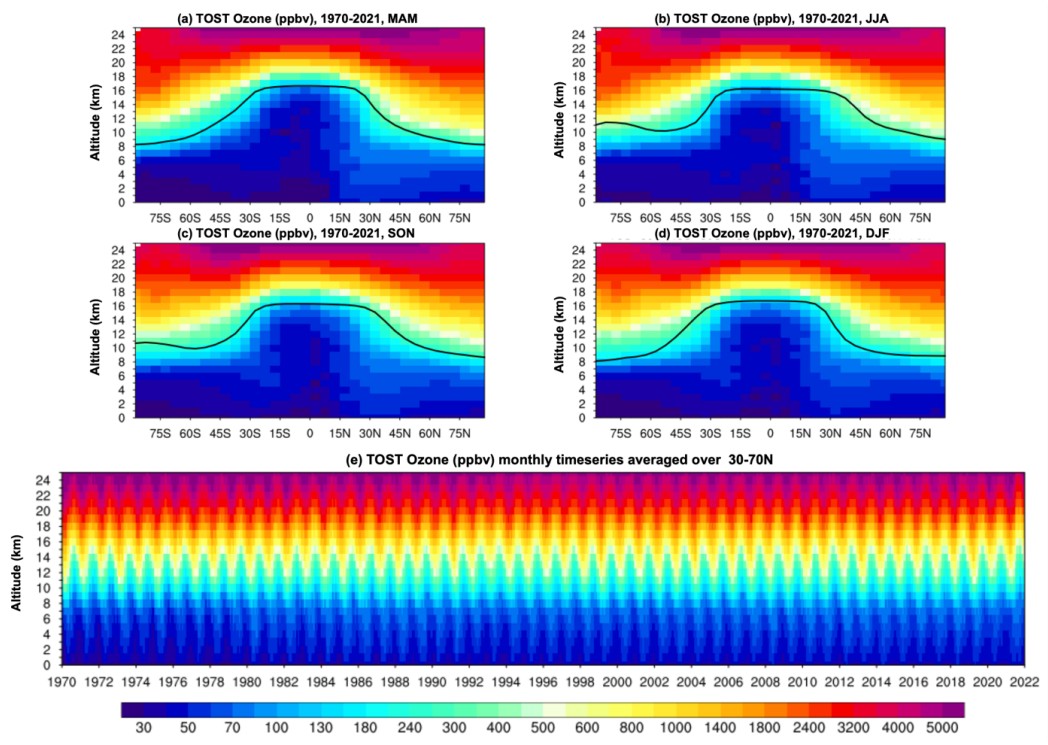



Figure 12. (a-d) The latitude-altitude distribution of TOST ozone averaged over 1970-2021 in each
season. The solid black lines represent the mean tropopause height over 1970-2021 in each season.
(e) time series of the monthly mean TOST ozone over 30-70°N at each altitude level from 1970 to

1032    2021.





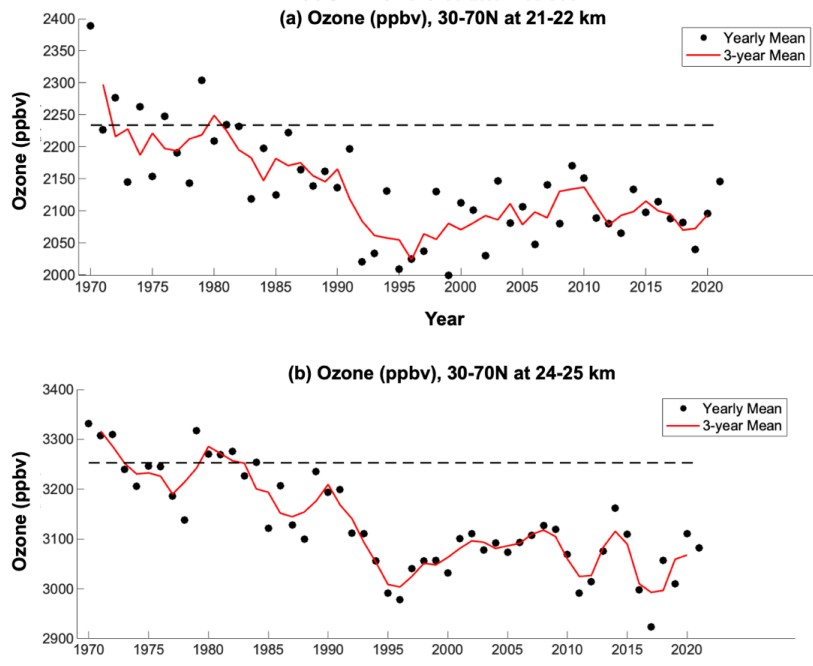

Figure 13. TOST time series of the annual mean ozone mixing ratios averaged over 30-70°N over 21-22 km altitude (a) and 24-25 km altitude (b). The black dots represent the annual mean ozone concentrations from the area-weighted average of the gridpoints over 30-70°N with ozone data throughout 1970-2021. The red line is the 3-year running mean. The black dashed line indicates the average ozone concentrations in the 1970s.