# Peer review of "An improved Trajectory-mapped Ozonesonde dataset for the Stratosphere and"

_EGUsphere, 2024_

## Community Comment (CC1)

Comments by Owen R. Cooper (TOAR Scientific Coordinator of the Community Special Issue) on:

**An improved Trajectory-mapped Ozonesonde dataset for the Stratosphere and Troposphere (TOST): update, validation and applications**

Zhou Zang, Jane Liu, David Tarasick, Omid Moeini, Jianchun Bian, Jinqiang Zhang, Anne M. Thompson, Roeland Van Malderen, Herman G. J. Smit, Ryan M. Stauffer, Bryan J. Johnson, and Debra E. Kollonige

EGUsphere [preprint], https://doi.org/10.5194/egusphere-2024-800, 2024.
Discussion started: 02 April 2024;  Discussion closes 6 June, 2024

This review is by Owen Cooper, TOAR Scientific Coordinator of the TOAR-II Community Special Issue. I, or a member of the TOAR-II Steering Committee, will post comments on all papers submitted to the TOAR-II Community Special Issue, which is an inter-journal special issue accommodating submissions to six Copernicus journals:  ACP (lead journal), AMT, GMD, ESSD, ASCMO and BG. The primary purpose of these reviews is to identify any discrepancies across the TOAR-II submissions, and to allow the author teams time to address the discrepancies.  Additional comments may be included with the reviews. While O. Cooper and members of the TOAR-II Steering Committee may post open comments on papers submitted to the TOAR-II Community Special Issue, they are not involved with the decision to accept or reject a paper for publication, which is entirely handled by the journal's editorial team.

**General Comments:**

TOAR-II has produced two guidance documents to help authors develop their manuscripts so that results can be consistently compared across the wide range of studies that will be written for the TOAR-II Community Special Issue.  Both guidance documents can be found on the TOAR-II webpage: https://igacproject.org/activities/TOAR/TOAR-II

*The TOAR-II Community Special Issue Guidelines*:   In the spirit of collaboration and to allow TOAR-II findings to be directly comparable across publications, the TOAR-II Steering Committee has issued this set of guidelines regarding style, units, plotting scales, regional and tropospheric column comparisons, tropopause definitions and best statistical practices.

*Guidance note on best statistical practices for TOAR analyses*:  The aim of this guidance note is to provide recommendations on best statistical practices and to ensure consistent communication of statistical analysis and associated uncertainty across TOAR publications. The scope includes approaches for reporting trends, a discussion of strengths and weaknesses of commonly used techniques, and calibrated language for the communication of uncertainty. Table 3 of the TOAR-II statistical guidelines provides calibrated language for describing trends and uncertainty, similar to the approach of IPCC, which allows trends to be discussed without having to use the problematic expression, "statistically significant".

**Specific Comments:**

A very important topic regarding detection of ozone trends in the troposphere is sampling frequency. Papers going back to the late 1980s have shown that low sampling frequencies (e.g. once per week ozone profiles) often fail to provide accurate monthly mean ozone values or reliable trends (Prinn, 1988; Logan, 1999; Cooper et al., 2010; Saunois et al., 2012; Chang et al., 2020; Chang et al., 2024). The modelling community is aware of this challenge (Lin et al., 2015; Barnes et al., 2016; Fiore et al., 2022) and they need long-term ozone observations with high sampling frequencies (greater than 3 times per week, if possible). The TOST product can help as it basically merges ozone observations on the regional scale, according to transport pathways, rather than through simple averaging across a pre-defined region. It would be very helpful to the modelling community if you could create a map that indicates the regions with the highest sampling frequencies, for example, areas with three or more observations per week, and regions with 5 or more observations per week.

The panels in Figure 9 are similar to my suggestion, but I'm not sure how to interpret these plots. For example, Figure 9e shows a dark green square over Hilo, Hawaii, which seems to indicate more than 180 samples for the month of January during 2000-2009. If I divide 180 by 10 years, then I get 18 ozone samples in a month, or a sampling frequency of more than 4 times per week. Sondes are only launched from Hilo once per week, so the other samples must be due to observations associated with the forward and backward trajectories. Given that Hilo is in the middle of the Pacific Ocean, it is probably more than 4 days of transport time from the nearest ozonesonde site, and therefore any trajectory in the 5x5 grid cell above Hilo must be associated with a Hilo ozonesonde. If this is the case, then the samples in the 5x5 grid cell are not independent. The algorithm must be counting the same observation several times while the trajectory slowly traverses the 5x5 grid cell.

Is there a way for you to determine the number of independent ozone values in a 5x5 grid cell? For example, can a forward or backward trajectory from a single ozonesonde only be counted once if it falls within a particular grid cell? If you can then make a plot showing the number of independent observations within a grid cell, then it is easier to relate TOST to a sampling strategy of 1, 3 or 5 profiles per week.

Lines 44-51

This introductory paragraph focuses on stratospheric ozone, while the topic of the TOAR-II Community Special Issue is tropospheric ozone. It's fine to discuss stratospheric ozone, as it impacts the troposphere, but a brief summary of the importance of tropospheric ozone is needed, especially when stating the importance of ozone for climate, as most of the radiative forcing is in the troposphere. See Chapters 2, 6 and 7 of IPCC AR6 (Gulev et al., 2021; Szopa et al., 2021; Forster et al., 2021).

When presenting the findings from the updated TOST product the focus is on the stratosphere and there is no analysis regarding tropospheric trends. The TOST product was used by the first phase of TOAR to show that ozone has increased in both hemispheres from 1998 to 2012 (Gaudel et al. 2018). TOST was also used by IPCC AR6, and the 1998-2012 positive ozone trends are consistent with the IAGOS trends over a slightly longer period (1994-2016), as shown in Figure 2.8 below (Szopa et al., 2021). It would be helpful to provide updated tropospheric ozone trends based on TOST-v2. It would also be helpful to show the extent of the negative ozone anomalies in 2020 caused by the COVID-19 economic downturn, as previously reported by Steinbrecht et al. 2021 and Putero et al., 2023 (published in the TOAR-II Community Special Issue).

Line 26

When saying the dataset has been updated to the most recent decade (1970s-2010s) it gives the impression that the final year in the dataset is 2019, but the final year is actually 2021. Please just list the full range of the dataset using the first and final years.

Lines 52-60
When reviewing the availability of long-term ozone profile records, please also mention lidar records. The lidar record at Observatoire de Haute Provence in southeastern France began in 1991; while the annual ozone anomalies from the lidar and the co-located ozonesondes differ due to sampling differences, both show a similar long-term ozone increase in the free troposphere, in the range of 1-3 ppbv/decade for 1991-2020 (Ancellet et al., 2022). Similarly, the JPL Table Mountain lidar north of Los Angeles shows an increase of 1 ppbv/decade for 2000-2023, as shown in the updated figure below (produced by Kai-Lan Chang using the method described by Chang et al., 2023). Since 2018 the Table Mountain lidar has a very high sampling frequency of 4-5 times per week. It also shows the decrease in ozone levels in 2020, associated with the COVID-19 pandemic.

Line 61
In addition to providing ozone retrievals for the stratosphere and troposphere, satellites also provide total column ozone retrievals.

Line 62
Satellite products can provide ozone retrievals for the lower, mid- and upper troposphere, with varying degrees of sensitivity, not just for the 6-10 km range (see section 3.3 of Gaudel et al., 2018)

Lines 68-72
A new area of global modelling involves the production of chemical reanalyses, which assimilate satellite data, to improve the quantification of tropospheric ozone, e.g. Miyazaki et al., 2020a,b; Colombi et al., 2021.

Line 130
The Data and Methods section needs to state how the tropopause is defined, as the product is provided in terms of both the troposphere and stratosphere. If a forward or backward trajectory begins in the troposphere and the final location of the trajectory particle, after 4 days, is above the tropopause, is this ozone observation categorized as being in the troposphere, or stratosphere?

Line 249-251
How were the IAGOS data averaged temporally? Into monthly means? What is the horizontal resolution? 5x5 degrees? How many airports were used? Did you use just the vertical profiles, or also the cruise level data? Do you have a data availability threshold? For example, do you require at least 4 profiles in a month to produce a monthly mean?

Line 540
The *Guidance note on best statistical practices for TOAR analyses* (described above) asks for all trends to be reported with 95% confidence intervals and p-values, and in units of ppbv decade$^{-1}$. In the submitted manuscript trends are only reported for the stratosphere and in units of ppbv year$^{-1}$. If ppbv year$^{-1}$ is the standard unit for reporting ozone trends in the stratosphere, then please retain this unit, otherwise please follow the TOAR guideline.

Figure S3
Why compare IAGOS and TOST over the range 50-150 ppb which includes tropospheric and stratospheric samples? If a monthly mean value for IAGOS observations is 100 ppb then it is very likely composed of both tropospheric samples (less than 100 ppb) and stratospheric samples (greater than 100 ppb). According to Figure S3d an IAGOS monthly mean of 100 ppb can correspond to a TOST value anywhere from 50 ppbv (mostly tropospheric samples) to 150 ppbv (mostly stratospheric samples). Clearly these two data sets are not sampling the same air masses and this is not an apples-to-apples comparison, so I don't see the value in these correlation plots.

Section 4.2

Every year, stratospheric ozone trends and variability are updated in the Global Climate chapter of the State of the Climate reports.  The most recent edition (Dunn et al., 2023) provides an update through the end of 2022. In particular, Figure 2.64 compares several products and shows stratospheric ozone levels at 22 km for the latitude band 35N-60N, similar to your Figure 13.  The SWOOSH product (Davis et al., 2016) is a combined satellite product, bias corrected against ozonesonde observations and provides global coverage.  How does TOST compare to these other products, and does TOST provide any new information?

[Figure]

Curtain plot of the Table Mountain lidar record, 2000-2023 (data provided by Thierry Leblanc, JPL; method described by Chang et al., 2023). The decrease in 2020 has been attributed to the impact of the COVID-19 economic downturn (Steinbrecht et al., 2021).

[Figure]

Figure 2.8 in IPCC, 2021:
Chapter 2. In: Climate Change 2021: The Physical Science Basis. Contribution of Working Group I to the Sixth Assessment Report of the Intergovernmental Panel on Climate Change
doi: 10.1017/9781009157896.004

**References:**

Ancellet, G., Godin-Beekmann, S., Smit, H. G. J., Stauffer, R. M., Van Malderen, R., Bodichon, R., and Pazmiño, A.: Homogenization of the Observatoire de Haute Provence electrochemical concentration cell (ECC) ozonesonde data record: comparison with lidar and satellite observations, Atmos. Meas. Tech., 15, 3105–3120, https://doi.org/10.5194/amt-15-3105-2022, 2022.

Barnes, E. A., Fiore, A. M., and Horowitz, L. W.: Detection of trends in surface ozone in the presence of climate variability, Journal of Geophysical Research: Atmospheres, 121, 6112–6129, https://doi.org/10.1002/2015JD024397, 2016.

Chang, K.-L., Cooper, O. R., Gaudel, A., Petropavlovskikh, I., and Thouret, V.: Statistical regularization for trend detection: An integrated approach for detecting long-term trends from sparse tropospheric ozone profiles, Atmospheric Chemistry and Physics, 20, 9915–9938, https://doi.org/10.5194/acp-20-9915-2020, 2020.

Chang, K.-L., Cooper, O. R., Rodriguez, G., Iraci, L. T., Yates, E. L., Johnson, M. S., et al. (2023). Diverging ozone trends above western North America: Boundary layer decreases versus free tropospheric increases. Journal of Geophysical Research: Atmospheres, 128, e2022JD038090. https://doi.org/10.1029/2022JD038090

Chang, K.-L., Cooper, O. R., Gaudel, A., Petropavlovskikh, I., Effertz, P., Morris, G., and McDonald, B. C.: Technical note: Challenges of detecting free tropospheric ozone trends in a sparsely sampled environment, EGUsphere [preprint], https://doi.org/10.5194/egusphere-2023-2739, 2024.

Colombi, N., Miyazaki, K., Bowman, K.W., Neu, J.L. and Jacob, D.J., 2021. A new methodology for inferring surface ozone from multispectral satellite measurements. Environmental Research Letters, 16(10), p.105005.

Cooper, O. R., Parrish, D. D., Stohl, A., Trainer, M., Nédélec, P., Thouret, V., Cammas, J.-P., Oltmans, S., Johnson, B. J., Tarasick, D., Leblanc, T., McDermid, I. S., Jaffe, D. A., Gao, R., Stith, J., Ryerson, T., Aikin, K., Campos, T., Weinheimer, A., and Avery, M. A.: Increasing springtime ozone mixing ratios in the free troposphere over western North America, Nature, 463, 344–348, https://doi.org/10.1038/nature08708, 2010.

Davis, S. M., Rosenlof, K. H., Hassler, B., Hurst, D. F., Read, W. G., Vömel, H., Selkirk, H., Fujiwara, M., and Damadeo, R.: The Stratospheric Water and Ozone Satellite Homogenized (SWOOSH) database: a long-term database for climate studies, Earth Syst. Sci. Data, 8, 461–490, https://doi.org/10.5194/essd-8-461-2016, 2016.

Dunn, R. J. H., J. B Miller, K. M. Willett, and N. Gobron, Eds., 2023: Global Climate [in "State of the Climate in 2022"]. Bull. Amer. Meteor. Soc., 104 (9), S11–S145, https://doi.org/10.1175/BAMS-D-23-0090.1.

Fiore, A.M., et al. (2022), Understanding recent tropospheric ozone trends in the context of large internal variability: A new perspective from chemistry-climate model ensembles, Environmental Research: Climate, https://doi.org/10.1088/2752-5295/ac9cc2

Forster, P., T. Storelvmo, K. Armour, W. Collins, J.-L. Dufresne, D. Frame, D.J. Lunt, T. Mauritsen, M.D. Palmer, M. Watanabe, M. Wild, and H. Zhang, 2021: The Earth's Energy Budget, Climate Feedbacks, and Climate Sensitivity. In Climate Change 2021: The Physical Science Basis. Contribution of Working Group I to the Sixth Assessment Report of the Intergovernmental Panel on Climate Change [Masson-Delmotte, V., P. Zhai, A. Pirani, S.L. Connors, C. Péan, S. Berger, N. Caud, Y. Chen, L. Goldfarb, M.I. Gomis, M. Huang, K. Leitzell, E. Lonnoy, J.B.R. Matthews, T.K. Maycock, T. Waterfield, O. Yelekçi, R. Yu, and B. Zhou (eds.)]. Cambridge University Press, Cambridge, United Kingdom and New York, NY, USA, pp. 923–1054, doi:10.1017/9781009157896.009.

Gulev, S.K., P.W. Thorne, J. Ahn, F.J. Dentener, C.M. Domingues, S. Gerland, D. Gong, D.S. Kaufman, H.C. Nnamchi, J. Quaas, J.A. Rivera, S. Sathyendranath, S.L. Smith, B. Trewin, K. von Schuckmann, and

R.S. Vose, 2021: Changing State of the Climate System. In Climate Change 2021: The Physical Science Basis. Contribution of Working Group I to the Sixth Assessment Report of the Intergovernmental Panel on Climate Change [Masson-Delmotte, V., P. Zhai, A. Pirani, S.L. Connors, C. Péan, S. Berger, N. Caud, Y. Chen, L. Goldfarb, M.I. Gomis, M. Huang, K. Leitzell, E. Lonnoy, J.B.R. Matthews, T.K. Maycock, T. Waterfield, O. Yelekçi, R. Yu, and B. Zhou (eds.)]. Cambridge University Press, Cambridge, United Kingdom and New York, NY, USA, pp. 287–422, doi:10.1017/9781009157896.004

Lin, M., et al. (2015), Revisiting the evidence of increasing springtime ozone mixing ratios in the free troposphere over western North America, Geophys. Res. Lett., 42, doi:10.1002/2015GL065311

Logan, J. A.: An analysis of ozonesonde data for the troposphere: Recommendations for testing 3-D models and development of a gridded climatology for tropospheric ozone, Journal of Geophysical Research: Atmospheres, 104, 16 115–16 149, 1999.

Miyazaki et al. (2020a), Evaluation of a multi-model, multi-constituent assimilation framework for tropospheric chemical reanalysis Atmos. Chem. Phys. 20 931–67

Miyazaki K et al. (2020b), Updated tropospheric chemistry reanalysis and emission estimates, TCR-2, for 2005-2018, Earth Syst. Sci. Data 12 2223–59

Prinn, R. G.: Toward an improved global network for determination of tropospheric ozone climatology and trends, Journal of Atmospheric Chemistry, 6, 281–298, https://doi.org/10.1007/BF00053861, 1988.

Putero, D., et al. (2023), Fingerprints of the COVID-19 economic downturn and recovery on ozone anomalies at high-elevation sites in North America and western Europe, Atmos. Chem. Phys., 23, 15693–15709, https://doi.org/10.5194/acp-23-15693-2023

Saunois, M., Emmons, L., Lamarque, J.-F., Tilmes, S., Wespes, C., Thouret, V., and Schultz, M.: Impact of sampling frequency in the analysis of tropospheric ozone observations, Atmospheric Chemistry and Physics, 12, 6757–6773, https://doi.org/10.5194/acp-12-6757-2012, 2012.

Steinbrecht, W. et al.(2021), COVID-19 Crisis Reduces Free Tropospheric Ozone Across the Northern Hemisphere, Geophysical Research Letters, 48, e2020GL091987. https://doi.org/10.1029/2020GL091987

Szopa, S., V. Naik, B. Adhikary, P. Artaxo, T. Berntsen, W.D. Collins, S. Fuzzi, L. Gallardo, A. Kiendler-Scharr, Z. Klimont, H. Liao, N. Unger, and P. Zanis, 2021: Short-Lived Climate Forcers. In Climate Change 2021: The Physical Science Basis. Contribution of Working Group I to the Sixth Assessment Report of the Intergovernmental Panel on Climate Change [Masson-Delmotte, V., P. Zhai, A. Pirani, S.L. Connors, C. Péan, S. Berger, N. Caud, Y. Chen, L. Goldfarb, M.I. Gomis, M. Huang, K. Leitzell, E. Lonnoy, J.B.R. Matthews, T.K. Maycock, T. Waterfield, O. Yelekçi, R. Yu, and B. Zhou (eds.)]. Cambridge University Press, Cambridge, United Kingdom and New York, NY, USA, pp. 817–922, doi:10.1017/9781009157896.008

---

## Author Comment (AC1)

**Review 1**

The paper describes an updated and improved version of the Trajectory Mapped Ozonesonde Dataset (TOST), which provides gridded ozone profile data from the 1970s until 2020. Overall this appears to be a good dataset. The paper is overall OK and suitable for ACP (or even better ESSD?). However, there are a number of issues that should be addressed before publication.

The paper is very long and contains a lot of redundant information in text and plots. Further down I suggest a number of way to simplify Figures. I strongly suggest to also shorten the corresponding text and to shorten and focus the conclusions section.

*Response: Thanks for taking time to review our paper and for your helpful suggestions and comments.*

1. It appears the TOST data set uses a 5°x 5°x 1km latitude x longitude x altitude grid (e.g. lines 26, 183, 184). However, it is not really clear what the provided time coordinate is. From lines 189 and 190 it appears that one time coordinate might be 12 monthly means, for each of the 5 decades 1970 to 1979, 1980 to 1989, ..., 2010 to 2019. Another time coordinate seems to be 52 annual means for each of the years 1970 to 2021. This should be clarified in a few places, especially in Abstract and Conclusions. Also, it begs the question, why the data-set is not simply provided as 12 monthly means for each of the 52 years.

*Response: Thanks for the points. The data are provided at three temporal resolutions: seasonal, annual mean and decadal-monthly mean. This information is now explicitly provided in Abstract, Conclusions, and text*

In Line 24-27 in Abstract:

*"Here, the seasonal, annual and decadal-monthly Trajectory-mapped Ozonesonde dataset for the Stratosphere and Troposphere (TOST) ozone climatology is improved and updated from 1970-2021 on a grid of 5° × 5° × 1 km (latitude, longitude, and altitude) from the surface to 26 km by geometric and pressure coordinates"*

In Line 660-663 in Conclusions:

*"Similar to TOST-v1, the ozone in each season, in each year (1970-2021) and in each month of a decade (January to December from the 1970s to 2010s) are provided in 3-dimensional grids of 5° × 5° × 1 km (latitude, longitude, and altitude)."*

We also summarize the coordinate systems, starting levels, temporal resolution, and data types for TOST data in Method (Line 239-244) and Table 1:

*"Based on this mapping, TOST-v2 was generated at 26 altitude levels in two altitude coordinates (by geometric levels and pressure levels), from two altitude starting levels (altitude above sea level and altitude above ground level), for three temporal resolutions (in the seasonal mean for each year, the annual means for each year from 1970 to 2021, and monthly means for each decade from the 1970s to the 2010s) and with three types of data fields (trop-strat, troposphere-only and stratosphere-only) for users' convenience (Table 1)."*

*Table 1. The classifications and descriptions for TOST data on coordinate systems, starting levels, temporal resolution, and data types.*

| Main Classifier | Type | Description |
|---|---|---|
| 1. Coordinates | Geometric | Altitude coordinates are 1, 2, …, 25, and 26 km at 1- |

| | | km vertical resolution. |
|---|---|---|
| | Pressure | Altitude coordinates are 950, 850, 750, 650, 550, 450, 400, 350, 300, 250, 225, 200, 175, 150, 125, 100, 90, 80, 70, 60, 50, 40, 35, 30, 25, 20 hPa. |
| 2. Starting levels | Sea-level | Data start at the altitude of the sea surface. Values for levels below the ground surface are set to null. |
| | Ground-level | Data start at the altitude of the ground surface. |
| 3. Temporal resolutions | Seasonal | Data are the mean for each season of the year (1970–2021). |
| | Annual | Data are the mean for each year (from 1970-2021). Each grid requires at least one value per season to be included in the annual data. |
| | Decadal-monthly | Data are the mean for each month of a decade (from the 1970s to 2010s). |
| 4. Ozonesonde data used | Trop_Strat | Data are based on ozonesonde profiles from both the troposphere and stratosphere. |
| | Troposphere-only | Data are based on ozonesonde profiles only from the troposphere. |
| | Stratosphere-only | Data are based on ozonesonde profiles only from the stratosphere. |
| 5. Ozone variations | Mean | The mean trajectories for each grid point over a period (a month, a year, or a month of a decade). |
| | 25th, 50th and 75th percentiles | The 25th, 50th and 75th percentiles of the trajectories for each grid point over a period (a month, a year, or a month of a decade). |
| 6. Supplement data | Smoothed data | Smoothed ozone fields by fitting the maps at each level with a linear combination of spherical functions (only for decadal-monthly and annual data). |
| | N | The number of trajectories passing through the grid cell. |
| | Number of independent samples | The trajectory was counted only once in a grid cell when the trajectory passes that cell regardless of how long the trajectory stays in that cell. |
| | Var | The variability of the trajectories values in each grid cell. |
| | SE | The ratio of the Var to the square root of the number of trajectories in each grid cell. |
| | SD | The ratio of the Var to the mean of trajectories values in each grid cell. |
| | SE/Mean | The ratio of the SE to the mean of trajectories values in each grid cell. |

The reason we could not provide monthly-mean data is because, despite the trajectory filling, monthly-mean data still have large gaps. Therefore, to increase the data availability, we provide the seasonal, annual and decadal-monthly mean data, which can be used for spatial analysis.

It is worth noting that all validation is carried out at monthly time step at individual stations or by regional mean.

2. To me, the paper contains way to many similar plots and panels. This makes it very hard for a reader. If there is no significant difference between seasons, decades, ... just show one plot / panel. See e.g. my comment on Fig. 5 below. Additional plots could go to the supplement, but even there: If there is no significant difference between seasons, decades, ... just show one plot / panel. The goal of the paper should be to clearly bring out the major messages, not to overwhelm and confuse the reader with redundant information.
Response: Thanks for the suggestion. We have revised Figures 5, 6 and 9 to clarify and simplify the contents. Please see more details in the following responses.

3. I am quite confused by the various relative and absolute measures used for differences in the validation part of the paper. Sometimes the authors seem to use mean relative difference (RD), sometimes bias (=absolute mean difference?) , sometimes root mean square differences (RMS, absolute or relative?), sometimes root mean square differences of the mean (RMS/sqrt(N), absolute or relative?). I think this should be clarified, and if at all possible simplified and unified. Response: Thanks! Apart from correlation coefficient (R) and linear fitting coefficient, we used relative difference (RD) and normalized root mean square (NRMS) to represent the **relative** difference between the two compared data, and used bias and root mean square (RMS) to show the **absolute** difference between the two compared data.

The relative difference allows comparing uncertainties and accuracies for TOST ozone estimations at different altitudes where ozone concentrations vary greatly. We have added the detailed equations for these metrics in Section S1 in the supplement file:
*"Multiple metrics were used to indicate the agreement and differences between the TOST (use y here) and ozonesonde/aircraft data (use x here).*

*1. Correlation Coefficient (R, unitless):*
$$R = \frac{\Sigma(x_i - \bar{x})(y_i - \bar{y})}{\sqrt{\Sigma(x_i - \bar{x})^2 \, \Sigma(y_i - \bar{y})^2}}, \text{ where } \bar{x} \text{ and } \bar{y} \text{ is the mean of the x and y variables, respectively.}$$
*2. Linear fitting coefficient (m, unitless), with the intercept set to 0:*
$$m = \frac{\Sigma(x_i y_i)}{\Sigma(x_i^2)}$$

*3. Bias (in ppb):*
$$Bias = \frac{1}{n}\sum(x_i - y_i)$$
*4. Relative Difference (RD, in %):*
$$RD = 100 \times \frac{Bias}{\bar{x}}, \text{ if comparing with ozonesonde data } x$$
$$RD = 100 \times \frac{Bias}{\frac{1}{2n}\Sigma(x_i + y_i)}, \text{ if comparing with satellite data } x$$
*5. Root Mean Square (RMS, in ppb):*
$$RMS = \sqrt{\frac{1}{n}\sum(x_i - y_i)^2}$$

*6. Normalized Root Mean Square (NRMS, in %):*

$$NRMS = 100 \times \frac{RMS}{\bar{x}}, \text{ if comparing with ozonesonde data } x$$

We have stated the metrics and their definitions in Method (Line 269-276):
*"Multiple metrics were used to indicate the agreement and differences between TOST and other data. We used correlation coefficient (R) to present the agreement of the two compared datasets, and linear fitting coefficient with the intercept set to 0 to show the overall tendency of overestimation/underestimation. We also used relative difference (RD) and normalized root mean square (NRMS) to represent the relative difference between the two compared data, and used bias and root mean square (RMS) to show the absolute difference between the two compared data. Details of the metrics can be found in Section S1."*

3.1 One such confusing example is Table S2, where I have no clue in what units the various quantities are given. I assume RMS is in ppbv, which is kind-of meaningless because ~400 ppbv would be a huge 400% uncertainty in the troposphere, and a reasonable 10% uncertainty in the stratosphere. I also assume that bias is in ppbv (absolute difference), and is essentially the same as RD (which seems to be relative difference in %). If relative and absolute difference ar given (RD and bias?), why are not also relative and absolute RMS given? In Figure 2, there is a sensible separation between tropospheric, stratospheric and intermediate ozone regimes. Why is that not done here in Table S2?
Response: Sorry for the confusion. We have now added units in the original Table S2 (now Table 3). We have used the "RMS and NRMS" in corresponding to the "RD and bias". Because the comparisons between two satellite data and TOST with ozonesonde data are only for >16 km, there is no separation for the three ozone regimes for Table S2 (now Table S3).

4. Line 363 and following: What is RMSE? Not defined. I assume it is root mean square error. How is that different from RMS difference?
Response: Yes, it is RMS, sorry for the confusion. Only RMS and NRMS are used in this revision. Please see our response to Comment 3.

5. Line 460, 461: What is NRMSE? Needs to be defined. It seems to be the same as relative root mean square error / difference
Response: NRMS is defined as the root mean square divided by the mean value of the variable. Please see our responses to Comment 3 for details.
Why is it not in %?
Response: Yes, NRMS should be in %.
In most other places relative differences and relative uncertainties are in % (and absolute ones in ppbv). Please define better and make consistent, e.g. always give RD and RMS in % and ppbv.
Response: We have unified the metrics and given RD and NRMS in % to indicate they are relative differences, and given bias and RMS in ppb to indicate they are absolute differences.

6. Figure 2: I find the vertical bars for R quite confusing. I would much prefer a third set of symbols / lines.
Response: For Figure 2, we have plotted the metrics separately to avoid confusion in this revision. To simplify the metrics, only R and RD are shown now.

[Figure]

*Figure 2. (a-c) Comparison of monthly average tropospheric ozone mixing ratios from ozonesondes (Sonde-Observed) and trajectory-derived TOST data (Traj-Derived) for the entire study period of ozone concentration at 0-50 ppb, 50-150 ppb and >150 ppb. Solid red lines represent the linear fitting line (with the intercept set to 0) and dashed black lines denote the 1:1 axis. N is the total number of data points, R is the correlation coefficient, Bias is the overall average difference in monthly mean values [Traj-Derived ozone - Sonde-Observed ozone, in ppb], RD is the relative difference in % [100 × (Traj-Derived ozone - Sonde-Observed ozone)/ Sonde-Observed ozone)], and RMS is the root mean square difference in ppb). Note that Traj-Derived ozone at each station is derived without input from the station itself; that is, Traj-Derived represents an ensemble of 141 separate computations of TOST, each one withholding a single validation station. (d-e) the R and RD between the Traj-Derived ozone and Sonde-Observed ozone by decade. The dashed line in (e) denotes where the RD is 0.*

I assume that each dot corresponds to one latitude-longitude-altitude grid-cell and one annual mean? Should probably be stated somewhere.
Response: Each dot in Figure 2a-c represents the monthly mean ozone value in one latitude-longitude-altitude grid cell. We have mentioned in the first paragraph of 3.1 in Line 333-335:
*"First, we show the overall comparison in monthly mean ozone profile between ozonesonde and trajectory-derived values without the inputs of the stations being tested (Traj-Derived), from all the existing stations at all altitude levels…"*
We also stated it again in Line 350-352:
*"Each dot in Figure 2a-c represents the paired ozone concentrations from the Traj-Derived and Sonde-Observed values in each month at each latitude-longitude-altitude grid-cell, and the color indicates the density of dots…"*

7. Figure 3: Why not also give numbers for the spread / width of the distributions, e.g. full-width at half maximum, or 1 standard deviation? I assume that the underlying data points are one latitude-longitude-altitude grid-cell and twelve calender months? Should probably be stated somewhere.

Response: Thanks for the good advice. We indicated the width of 1 standard deviation using thick red lines for the RD in each level and gave the value in red. Also, because the peak density values are not the focus of this plot, we deleted the values and kept only the points indicating where the density peaks.

[Figure]

*Figure 3. The relative difference (RD) of the monthly ozone mixing ratios between ozonesonde and Traj-Derived data by altitude in the 1970s, 1980s, 1990s, 2000s and 2010s, respectively. The frequency distribution of RD at every other altitudes is shown (y-axis: frequency in %, x-axis: RD in %), with the colors denoting the 4 quartiles of RD. The dashed line indicates zero difference in RD. The blue dot represents the maximum frequency. The thick red lines denotes the width of distribution at 25-75%-ile, with the corresponding width of the distribution value in red.*

The data to calculate the RD distribution is from the monthly mean ozone data between ozonesonde and Traj-Derived data from all the existing stations at selected altitude levels. We have emphasized the data points for RD distribution in Line 387-389:
*"The RD distributions are based on the monthly ozone concentration difference between the actual ozonesonde and Traj-Derived data from all the existing stations at the corresponding altitude level and decade."*

8. Figure 5: I don't see any clear or significant differences between the top four panels, or between the bottom four panels. Therefore, I strongly suggest to just have one panel showing SAGE - TOST (all seasons, years), and one panel showing MLS - TOST (all seasons, years). It would, however, be helpful to also plot the relative RMS differences.
9. Figure 6: There is a lot of redundancy between Fig. 6 and Fig. 5. The single profile panels of Fig. 6 contain more or less the same information as Fig. 5 (especially if my suggested reduction is done). The main additional information in Fig. 6 is the seasonal variation (which is clearly

visible for MLS). Maybe there is no need for Fig. 5, or the single profile panels of Fig 6. could be dropped?
Response: Thanks. Figure 5 is now removed. Please see our responses to Comment 6.

10. Figure 7, Figure S3: Again, I don't see the need for four panels, as I don't see a significant difference between the panels. On the other hand the split between < 50 ppbv and 50 to 150 ppbv seems very artificial here. It seems to me that just one panel that includes all data from 0 to 150 ppbv would be enough and more sensible here.
Response: Thanks for the suggestion. The separations of <50 ppb and 50-150 ppb here are to see the comparisons of IAGOS and TOST data in the lower and upper troposphere. However, since ozone of 100-150 ppb can have a tropospheric or lower stratospheric source, the latter comparison is less meaningful. To make the comparison apples-to-apples, we only keep Figure 7 (the comparison of <50 ppb ozone samples) to make sure both IAGOs and TOST ozone are from tropospheric air, which is also the purpose of this figure: to compare the tropospheric ozone from TOST to another broadly trusted tropospheric ozone data (IAGOs).

11. Line 441 and following: SE/mean is that not simply the relative RMS/sqrt(N) (in %). Another example where a more consistent nomenclature and use of relative and absolute differences would be helpful.
Response: Thanks for your points. SE/Mean here is the ratio of the standard error to the mean, while standard error is standard deviation/sqrt(N). Therefore, SE/Mean is not the RMS/sqrt(N) (please see the definition of RMS in our response to Comment 3). SE indicates a confidence interval for the mean averaged over all trajectories. We use "SE/Mean" for Figure 9 to provide a relative measure of the SE, to avoid any confusion due to the magnitude differences in ozone concentrations at different altitudes.

12. Figure 9: unless there is a large and significant seasonal variation: two rows might be enough. However, I would like to see a third column with relative RMS (in %, without the 1/sqrt(N)). I guess this third column would carry comparable information as Fig. 10?
Response: Thanks for the suggestion. The seasonal variation here is to show the uncertainties of TOST in different seasons, which is clear at the 3-4km altitude level that the warm season has higher SE/Mean than the cold season. Also SE/Mean varies less with season in the stratosphere than in the troposphere. Therefore, we still keep the seasonal variations in the figure.
If without the 1/sqrt(N), we present the coefficient of variance (CV, in %) here (Figure R1), which is calculated as the ratio of standard deviation to the mean value. Compared to SE/Mean, CV has a relatively larger value than SE/Mean, as 1/sqrt(N) is not included. For the reasons described in our responses to Comment 11, we have kept SE/mean in the figure.

[Figure]

Figure R1. Global distribution of the coefficient of variance (CV, in %) for January and July 2000s at 3-4 km (a and b) and 19-20 km (c and d).

These RMS numbers should be compared with estimates of ozone sonde uncertainty, e.g. those given be Tarasick et al. 2016, 2021.

Response: Note that the calculated RMS is a bias, while the "uncertainty" used by Tarasick et al. (2016, 2021) is random uncertainty from sources in the sonde instrument and measurement. We use the differences between Traj-Derived ozone and ozonesonde data to estimate the uncertainty in the TOST product, This is substantially larger than the (order ~5%) sonde measurement uncertainties discussed in Tarasick et al. (2016, 2021).

13. Figure 10: Would not Fig. 10 and this entire uncertainty discussion (section 3.5 and Figs. 9 and 10) fit much more logically directly after Figs. 3 and 4 and section 3.1, which also compares Traj-Derived with Sonde??

Response: Each panel in Figure 10 is meant to investigate how NRMS changes with altitudes, seasons, latitude zones and decades to give readers a clear view of where and when the uncertainty of TOST could be higher. This is not only the comparison between Traj-Derived and Sonde-observed ozone, but also an important caveat for users of TOST to know where and when the data would have higher uncertainties. Therefore, we hoped that putting this uncertainty discussion after the comparisons and validations of TOST data would serve as a good summary and caveats for the first part of this paper (validations and comparisons of TOST).

Is NRMSE not the same as relative RMS? Should it not also be given in %.

Response: Yes, we have unified this metric as "NRMS" in the manuscript, and the unit of NRMS is given in %. Please see our responses in detail to Comment 3.

Should panel a.) not also have altitude on the vertical coordinate, like all the other plots?
Response: To make the panels consistent (NRMS on the vertical coordinate and altitudes/seasons/latitude zones/decades on the horizontal coordinate), it is better to keep the altitudes on the horizontal coordinate.
These RMS numbers and the profile in panel a.) should be compared with estimates of ozone sonde uncertainty profiles, e.g. those given be Tarasick et al. 2016, 2021.
Response: As noted above, the "uncertainty" used by Tarasick et al. (2016, 2021) is random uncertainty from sources in the sonde instrument and measurement. These RMS numbers are substantially larger than the (order ~5%) sonde measurement uncertainties discussed in Tarasick et al. (2016, 2021).

14. Lines 503 to 505: This is important and needs to appear prominently also in the conclusions, and in the introduction (e.g. after line 91). We don't need another "tropical ozone hole" paper and consequent rebuttal like Chipperfield et al. 2022.
Response: Thanks for the suggestion. We have emphasized this incorrect use of TOST in the introduction in Line 125-128:
*"For users' convenience, the remaining gaps after trajectory mapping were further filled with a linear combination of spherical functions and provided as "smoothed" data in TOST-v1. Yet, the smoothed data should be used with caution; otherwise, misinterpretation of the smoothed data can be problematic (Chipperfield et al., 2022)."*
And in the conclusions in Line 706-708:
*"In addition, the smoothed dataset should be used for quantitative analysis with great caution, as it has not been quantitatively evaluated in any way."*

15. Line 79: should be "lower stratosphere". Above 25 km the lifetime of ozone becomes shorter.
Response: Revised. Thank you.

16. Line 263: should "tropospheric" not be deleted here? Otherwise, why not do stratospheric as well here?
Response: Deleted. Thank you.

17. Line 343: I don't see a comparable or better performance of MLS here, unless you mean smaller RMS / error bars, which are barely visible. In this context, see my suggestion above for Fig. 3, to add the RMS profiles to the plots, and to reduce the number of panels.
Response: Thanks for you correction. Yes, TOST has a comparable or better performance than MLS. Revised.

18. Line 390: 3d should proably be 7d
Response: Revised. Thank you.

---

## Author Comment (AC2)

**Review 2 by Michael Prather**

You already have some excellent suggestions from RC1. My issues mainly focus on the methodology and future of these ozone data sets.

This paper documents and presents a long-term global, gridded harmonized ozone data set for the troposphere and lower stratosphere (TOST) that is readily amenable for developing model metrics and studies of trends and interannual variability. It is very well written for the most part and will be a valuable addition to a broad community studying atmospheric chemistry and transport, global air pollution, and global change. The core datasets are the ozonesondes, and MOZAIC/IAGOS is used for validation – a great choice for well calibrated, highest resolution possible atmospheric composition measurements. The updated TOST-v2 is a great product. Yet, this is a disappointing paper in merely repeating the TOST-1 protocol without much thought as to the use of the data in modern models. At this point it needs to be published as is (with some minor noted corrections) but with the added recognition/recommendations of how to do it better.

Response: Thanks for taking the time to review our paper and for your helpful suggestions and comments.

1. To me, the obvious question here is: why not include the IAGOS data as a source for TOST? It seems like you are wasting a major resource by using it only for validation. I am not asking you to create TOST-3 for this paper, but at least you could discuss this at the beginning. Are there fundamental problems with this? or just too much work for now (that is OK).

Response: Thanks for this question. In fact, we started to do this quite some time ago and quickly became aware that there was a persistent bias between IAGOS and sonde measurements. The result of that is a recent study from Wang et al. (2024), which evaluated the agreement between IAGOS and ozonesonde data, resulting in all sonde types showing significant average biases with respect to IAGOS (higher by 5-10% than IAGOS), and the relative bias increasing modestly with altitude. This result also agrees well with our Figure 6, comparing IAGOS and TOST data (the relative difference is 5.84%). In addition, the time periods of IAGOS and ozonesonde data are not generally the same, so merging these two data types can introduce spurious jumps in timeseries, if the bias is not properly resolved.

At present, a similar trajectory-derived ozone climatology based on IAGOS data is now under construction in our group and will be published as an independent dataset like TOST.

2. Abst. "of 5º × 5º × 1 km (latitude, longitude, and altitude" sound nice but it is missing two important quantities: (1) is "altitude" really just altitude (km above the surface) or is it "pressure altitude" ? be specific (log p, or US STD atmos p like flight levels); (2) time is critical here, what is the resolution and method of averaging? I see in L164 that you used monthly averages, please state this up front.

Response: Thanks for the question, the altitude in the previous version is the geometric altitude above the surface, and now we have added the pressure altitude as suggested. The time is at two resolutions: annual mean and decadal-monthly mean.

Oh, now I see in L204 ("The resulting ozone fields are given in two altitude coordinates (altitude above sea level and altitude above ground level) for users' convenience") that you are using geometric altitude. This is really problematic since the altitude of the land surface depend

heavily on the resolution of the model you (and your users) are using. I think these are possibly the worst possible vertical coordinates you could use, especially for the 6-26 km region where the results are most reliable. The use of altitude requires one to know the temperature profile, which is seriously problematic since any model profile may NOT be what you use and there fore cannot be compared. If you are using a fixed T profile, then just provide the data set in pressure coordinates.

I think the data set must really be in pressure coordinates to be useful to any 3D model. This you can and should fix.

Response: Thanks for the good suggestion. We have added the TOST-v2 in pressure altitudes, including the altitude above sea level and altitude above ground level. We also mentioned this improvement in the Method (Line 369-371), Result (Line 1071-1072) and Conclusion (Line 1316-1388). In this study, we used geometric coordinates as the example for all the comparisons, validations and investigations.

We also described how we produced TOST in pressure altitudes in Method in Line 225-230:

*"To produce the mapping with pressure altitudes, we also averaged the 4-day backward and forward trajectories in bins of 5º latitude and 5º longitude for every month, using the pressure altitudes generated by HYSPLIT trajectories. The 26 pressure altitudes are 950, 850, 750, 650, 550, 450, 400, 350, 300, 250, 225, 200, 175, 150, 125, 100, 90, 80, 70, 60, 50, 40, 35, 30, 25, 20 hPa, which is determined and adjusted based on the ERA5 pressure coordinates and the 1-26km geometric altitudes."*

3. Overall big problem and opportunity – may be insurmountable, but should be recognized. Spatio-temporal averaging destroys the ozone structure anywhere near the tropopause. It is clear that this data set does not resolve tropopause ridges-troughs nor strat-trop folds – therefore the averaging of mole fraction ozone means that stratospheric ozone dominates the abundance well into the troposphere. You simply average the ozone mole fraction in your large cells over the month. It would be great to produce a more nuanced data set that considers the natural variability in ozone. Specifically, why not give 10-25-50-75-90 %iles, that way one can test the high resolution (no serious models are running % deg resolution anymore), high-frequency simulations. These statistics would help identify the frequency of strat-vs-trop, etc. and make model comparisons with the coarse resolution you use more informative. I think you should be more expansive in diagnosis.

Response: Thanks! Please note that we provide three versions of the TOST fields, based on the origin of the ozonesonde data (tropospheric or stratospheric, defined by the WMO definition from the measured ozonesonde temperature profile): "troposphere-only" and "stratosphere-only" and a combined "trop_strat" product. The last is indeed subject to the issues you note.

To study the variations of ozone, we have provided the standard deviation of ozone trajectories of each grid. As suggested, we now also added the annual and decadal-monthly 25-50-75 %-iles ozone mapping in TOST-v2. Because the number of trajectories could be limited, 10 and 90%-iles are not provided.

We also noted the meaning of providing the percentiles of ozone in Method at Line 244-246:

*"In TOST-v2, we also generate the corresponding datasets that show ozone variation at 3 percentile levels (25, 50 and 75%)."*

In Result at Line 554-555:

*"Furthermore, TOST-v2 provides additional information that shows ozone variations in 3 percentile levels (25, 50 and 75%)."*

And in Conclusion at Line 671-672:
*"In addition to the seasonal, annual or decadal-monthly means, the corresponding datasets for ozone variations at 3 percentile levels (25, 50 and 75%) are also provided."*

4. L61: The satellite data indeed have trouble with the troposphere (except with product involving cloud slicing or OMI-MLS as in Ziemke et al). I am even worried that MLS and SAGE may have difficulties in the UT/LS give the resolution you cite.
Response: Thanks for the question. Both SAGE and MLS are designed for measuring stratospheric ozone. It is recommended to use MLS ozone profiles only above 261 hPa (Livesey et al., 2022). We compared the MLS and SAGE profiles in Figure 6 using only >16km, which is even higher than the recommended altitude (~10 km) and should avoid comparing the too-large bias in the UT/LS area.

5. L77-79: The argument for ozone being inert for 4 days along the trajectory is reasonable for the UT/LS, but the out-of-date Jacob (1999) paper you use here is simply wrong for the lower troposphere. Look at the regions of intense ozone loss (>5 ppb/day) in the ATom transects (Prather, Guo, Zhu 2023, doi: 10.5194/essd-15-3299-2023) or the 3-5 day perturbation lifetime of surface ozone pollution in Prather & Zhu (2024, Lifetimes and timescales of tropospheric ozone, Elementa, doi: 10.1525/elementa.2023.00112). I do not think you can easily do anything (or even should do anything) about this for your TOST-2 product, but there should be a recognition of the potential error.
Response: Thanks for pointing out this out. Our results did show the bias from assuming a 4-day lifespan of ozone in the lower troposphere. For example, in Figure 3, the surface (boundary layer) ozone shows a positive bias of the median, in all decades, of up to 12%. In addition, in Figure 4, the larger discrepancies are shown near the planetary boundary layer due to the fact that a 4-day lifespan for ozone could be unreal for the lower troposphere. In the uncertain analysis, we emphasized that surface ozone could be more biased than other altitudes.
In this version, we have cited the study of Han et al. (2019) and Prather & Zhu (2024) in the introduction. In Han et al. (2019), the lifetime of ozone at the middle troposphere (500 hPa) and the surface is estimated to be >10 days and 1.1-11.3 days, respectively. Therefore, the extension of the 4-day lifespan for ozone is generally reasonable for generating the TOST data.
In future studies, we will improve the TOST in near-surface by using varied trajectory length for different altitudes of the atmosphere according to their mean lifespan.

6. L169: The new HYSPLIT may be numerically accurate but the NCAR/NCEP wind fields seem totally out of date – the vertical resolution (17 layers from 0 to 32 km = 2 km at best near the tropopause) can hardly resolve vertical motions in the UT/LS. Why not use more modern fields like ERA-5 or MERRA-2? It makes the paper look lazy, you updated the sondes, but just ran with the old parts of TOST-1. I know you cannot fix this, but it should be recognized as a problem (like the minimal use of IAGOS observations) that should be upgraded in TOST-3.
Response: Thanks for the suggestion. We agree that updated wind fields could improve TOST accuracy. While the 17 layer NCEP fields do lack vertical resolution, they were until fairly recently the only reanalysis dataset that offered consistency back to the 1960s, when our sonde data begin. Other NCEP data are for more recent years. MERRA starts in 1980. The ERA5 dataset would be an obvious improvement, but the effort involved in switching is considerable.

7. L272:  I was going to congratulate the authors on their correct use of nmol/mol as the measure of ozone abundance and then I hit the incorrect use of 'ppbv' ("RMS of 21.1 ppbv, and higher bias (2.9 ppbv) and").  The 'by volume' should have been scoured out of this community by now but many prominent colleagues continue to abuse this.  The 'volume' is not mole fraction since virial corrections would need to be applied, and most all measurements calibrate to dry air mole fraction.

Response: We meant ppb. Thanks for catching this.

8. L475:  You really should be comparing tropospheric O3 column (DU or mean ppb) with Ziemke et al's work.  The whole paper is well referenced within its limitations (noted above), but you simply must compare the features in Figures 8 and later with Ziemke's work.

Response: Thanks for the suggestion. This is not as simple as you might think, since many TOST columns are missing data at one or more levels. We are working on a better gap-filling estimation technique, and in an upcoming paper, we have compared the tropospheric O3 column with Ziemke et al's OMI data. Hope to submit it soon.

9. L555:  Again, note that this is monthly averaged.

Response: Thanks for pointing this out. The data is provided at three temporal resolutions: seasonal, annual and decadal-monthly mean, and we have emphasized this in the conclusion. The reason we could not provide monthly-mean data is that despite the trajectory filling, monthly-mean data still have large gaps. Therefore, to increase the data availability, we provide the seasonal, annual and decadal-monthly mean data, which can be used for spatial analysis.

**References**

Han, H., Liu, J., Yuan, H., Wang, T., Zhuang, B., and Zhang, X.: Foreign influences on tropospheric ozone over East Asia through global atmospheric transport, Atmospheric Chemistry and Physics, 19, 12495-12514, 2019.

Livesey, N., Read, W., Wagner, P., Froidevaux, L., Santee, M., Schwartz, M., Lambert, A., Millán Valle, L., Pumphrey, H., and Manney, G.: Earth Observing System (EOS) Aura Microwave Limb Sounder (MLS) version 5.0 x level 2 and 3 data quality and description document Version 5.0–1.1 a (Tech. Rep.), Jet Propulsion Laboratory, California Institute of Technology. Retrieved from https://mls. Jpl. Nasa. Gov/data/v5-0_data_quality_document. Pdf, 2022.

Prather, M. J. and Zhu, X.: Lifetimes and timescales of tropospheric ozone: Global metrics for climate change, human health, and crop/ecosystem research, Elementa: Science of the Anthropocene, 12, 2024.

---

## Author Comment (AC3)

**Review by Owen R. Cooper (TOAR Scientific Coordinator of the Community Special Issue)**

This review is by Owen Cooper, TOAR Scientific Coordinator of the TOAR-II Community Special Issue. I, or a member of the TOAR-II Steering Committee, will post comments on all papers submitted to the TOAR-II Community Special Issue, which is an inter-journal special issue accommodating submissions to six Copernicus journals: ACP (lead journal), AMT, GMD, ESSD, ASCMO and BG. The primary purpose of these reviews is to identify any discrepancies across the TOAR-II submissions, and to allow the author teams time to address the discrepancies. Additional comments may be included with the reviews. While O. Cooper and members of the TOAR-II Steering Committee may post open comments on papers submitted to the TOAR-II Community Special Issue, they are not involved with the decision to accept or reject a paper for publication, which is entirely handled by the journal's editorial team.

General Comments:
TOAR-II has produced two guidance documents to help authors develop their manuscripts so that results can be consistently compared across the wide range of studies that will be written for the TOAR- II Community Special Issue. Both guidance documents can be found on the TOAR-II webpage: https://igacproject.org/activities/TOAR/TOAR-II
The TOAR-II Community Special Issue Guidelines: In the spirit of collaboration and to allow TOAR-II findings to be directly comparable across publications, the TOAR-II Steering Committee has issued this set of guidelines regarding style, units, plotting scales, regional and tropospheric column comparisons, tropopause definitions and best statistical practices.
Guidance note on best statistical practices for TOAR analyses: The aim of this guidance note is to provide recommendations on best statistical practices and to ensure consistent communication of statistical analysis and associated uncertainty across TOAR publications. The scope includes approaches for reporting trends, a discussion of strengths and weaknesses of commonly used techniques, and calibrated language for the communication of uncertainty. Table 3 of the TOAR-II statistical guidelines provides calibrated language for describing trends and uncertainty, similar to the approach of IPCC, which allows trends to be discussed without having to use the problematic expression, "statistically significant".
Response: Thanks for taking the time to review our paper and for your helpful suggestions and comments.

Specific Comments:
1. A very important topic regarding detection of ozone trends in the troposphere is sampling frequency. Papers going back to the late 1980s have shown that low sampling frequencies (e.g. once per week ozone profiles) often fail to provide accurate monthly mean ozone values or reliable trends (Prinn, 1988; Logan, 1999; Cooper et al., 2010; Saunois et al., 2012; Chang et al., 2020; Chang et al., 2024). The modelling community is aware of this challenge (Lin et al., 2015; Barnes et al., 2016; Fiore et al., 2022) and they need long-term ozone observations with high sampling frequencies (greater than 3 times per week, if possible). The TOST product can help as it basically merges ozone observations on the regional scale, according to transport pathways, rather than through simple averaging across a pre-defined region. It would be very helpful to the modelling community if you could create a map that indicates the regions with the highest

sampling frequencies, for example, areas with three or more observations per week, and regions with 5 or more observations per week.

The panels in Figure 9 are similar to my suggestion, but I'm not sure how to interpret these plots. For example, Figure 9e shows a dark green square over Hilo, Hawaii, which seems to indicate more than 180 samples for the month of January during 2000-2009. If I divide 180 by 10 years, then I get 18 ozone samples in a month, or a sampling frequency of more than 4 times per week. Sondes are only launched from Hilo once per week, so the other samples must be due to observations associated with the forward and backward trajectories. Given that Hilo is in the middle of the Pacific Ocean, it is probably more than 4 days of transport time from the nearest ozonesonde site, and therefore any trajectory in the 5x5 grid cell above Hilo must be associated with a Hilo ozonesonde. If this is the case, then the samples in the 5x5 grid cell are not independent. The algorithm must be counting the same observation several times while the trajectory slowly traverses the 5x5 grid cell.

Is there a way for you to determine the number of independent ozone values in a 5x5 grid cell? For example, can a forward or backward trajectory from a single ozonesonde only be counted once if it falls within a particular grid cell? If you can then make a plot showing the number of independent observations within a grid cell, then it is easier to relate TOST to a sampling strategy of 1, 3 or 5 profiles per week.

Response: Thanks for the good suggestion. To determine the number of independent ozone values, we counted the forward and backward trajectories originated from an ozonesonde flying altitude only once if the trajectory falls within a particular grid cell regardless how long the trajectory stays in that cell. The number of independent samples are provided in the TOST data as well, named with a suffix of "*number_independent.asc".

The updated number of independent ozone values, for example, for January 2000s at 3-4 km and 19-20km is shown in Figure R1:

[Figure]

Figure R1. Global distribution of the number of independent samples for the annual mean ozone in 2000 at 3-4 km and 19-20 km.

In Hilo, the number of independent samples is 73 at 3-4 km, and 45 at 19-20 km in Jan, 2000s, which is about 4-7 samples per month (or 1-2 samples per week). The result shows that most of the samples are associated with the Hilo ozonesonde, yet still have some samples for trajectories outside the Hilo ozonesonde.

To confirm there are trajectories other than the Hilo ozonesonde to this station, we also calculated the number of independent samples for trajectories of 1-4 days. The 1-day trajectories will mostly reflect the number of samples from the ozonesonde stations, and we compare it with the 1-day trajectories generated only by the Hilo station. We found that Hilo station has 55 samples from 1-day trajectories at 3-4 km, which is the same as the 1-day trajectories generated only by Hilo station. The >1-day trajectories will mostly reflect the number of samples from other ozonesonde stations. For >1-day trajectories at 3-4 km, Hilo station has in total of 18 samples, indicating the influence of trajectories from other ozonesonde stations. Therefore, we believe the number of independent samples we calculated now is reasonable.

We also replaced the number of trajectory samples with the number of independent samples in Figure 7, so that the standard error now reflects the number of independent samples, and is correspondingly larger:

[Figure]

*Figure 7. (a-d) Global distribution of the SE/Mean (left panels, in %) for the decadal monthly mean ozone in January and July 2000s at 3-4 km (a and b) and 19-20 km (c and d). (e-h) the same as (a-d), but for the number of independent samples in each 5 × 5° bin.*

2. Lines 44-51

This introductory paragraph focuses on stratospheric ozone, while the topic of the TOAR-II Community Special Issue is tropospheric ozone. It's fine to discuss stratospheric ozone, as it impacts the troposphere, but a brief summary of the importance of tropospheric ozone is needed, especially when stating the importance of ozone for climate, as most of the radiative forcing is in

the troposphere. See Chapters 2, 6 and 7 of IPCC AR6 (Gulev et al., 2021; Szopa et al., 2021; Forster et al., 2021).

Response: Thanks for the suggestion. We have added the emphasis on the tropospheric ozone to the introduction in Line 48-52:

*"Ozone is an important oxidant photochemically linked to the hydroxyl radical in the troposphere, with detrimental effects on crop productivity, natural ecosystems and human health (Fleming et al., 2018; Mills et al., 2018; Harmens et al., 2018; Vicedo-Cabrera et al., 2019). Tropospheric ozone is the third largest greenhouse gas contributing to radiative forcing, particularly in the upper troposphere (Gulev et al., 2021; Szopa et al., 2021; Forster et al., 2021)."*

3. When presenting the findings from the updated TOST product the focus is on the stratosphere and there is no analysis regarding tropospheric trends. The TOST product was used by the first phase of TOAR to show that ozone has increased in both hemispheres from 1998 to 2012 (Gaudel et al. 2018). TOST was also used by IPCC AR6, and the 1998-2012 positive ozone trends are consistent with the IAGOS trends over a slightly longer period (1994-2016), as shown in Figure 2.8 below (Szopa et al., 2021). It would be helpful to provide updated tropospheric ozone trends based on TOST-v2. It would also be helpful to show the extent of the negative ozone anomalies in 2020 caused by the COVID-19 economic downturn, as previously reported by Steinbrecht et al. 2021 and Putero et al., 2023 (published in the TOAR-II Community Special Issue).

Response: Thanks for the good advice. Investigating the tropospheric ozone trend based on TOST-v2 is underway under project of "The Harmonization and Evaluation of Ground Based Instruments for Free Tropospheric Ozone Measurements (HEGIFTOM)" with careful comparison and estimates.

4. Line 26

When saying the dataset has been updated to the most recent decade (1970s-2010s) it gives the impression that the final year in the dataset is 2019, but the final year is actually 2021. Please just list the full range of the dataset using the first and final years.

Response: Revised. Thank you.

5. Lines 52-60

When reviewing the availability of long-term ozone profile records, please also mention lidar records. The lidar record at Observatoire de Haute Provence in southeastern France began in 1991; while the annual ozone anomalies from the lidar and the co-located ozonesondes differ due to sampling differences, both show a similar long-term ozone increase in the free troposphere, in the range of 1-3 ppbv/decade for 1991-2020 (Ancellet et al., 2022). Similarly, the JPL Table Mountain lidar north of Los Angeles shows an increase of 1 ppbv/decade for 2000-2023, as shown in the updated figure below (produced by Kai-Lan Chang using the method described by Chang et al., 2023). Since 2018 the Table Mountain lidar has a very high sampling frequency of 4-5 times per week. It also shows the decrease in ozone levels in 2020, associated with the COVID-19 pandemic.

Response: Thanks for the suggestion. We have added the lidar records for ozone profiles in Line 66-69:

*"In addition, lidar records also provide long-term tropospheric ozone profiles, such as the Observatoire de Haute Provence lidar and the Jet Propulsion Laboratory Table Mountain lidar (Ancellet and Beekmann, 1997; McDermid et al., 2002). However, the horizontal and temporal coverages of both ozonesondes and lidars are limited by the sparse distribution of the stations (less than 100 worldwide for ozonesondes; 9 lidars from the Tropospheric Ozone Lidar Network) and their low observation frequency (1-3 times/week for ozonesondes; 1-5 times/week for lidars) (McDermid et al., 2002; Liu et al., 2013a; Chouza et al., 2019; Ancellet et al., 2022)…"*

6. Line 61
In addition to providing ozone retrievals for the stratosphere and troposphere, satellites also provide total column ozone retrievals.
Response: Thanks for your point. We have revised the manuscript in Line 76-77:
*"Satellite data usually provide total column ozone retrievals that are not vertically resolved."*

7. Line 62
Satellite products can provide ozone retrievals for the lower, mid- and upper troposphere, with varying degrees of sensitivity, not just for the 6-10 km range (see section 3.3 of Gaudel et al., 2018)
Response: Thanks for the correction. We rephrased the disadvantages of satellite in observing tropospheric ozone profiles in Line 75-87:
*"However, it is still challenging to retrieve tropospheric ozone through the large stratospheric ozone burden (Bhartia, 2002). Satellite data usually provide total column ozone retrievals that are not vertically resolved. The satellite ozone profiles have limited sensitivity to fine-scale atmospheric structures and the sensitivity decreases strongly toward the surface (Liu et al., 2010; Keppens et al., 2015). The direct retrieval from nadir-viewing instruments typically provides 1 independent point in the troposphere (Tarasick et al., 2019b). Large retrieval errors occur when retrieval sensitivity is low, as the solution relies heavily on the a priori (Keppens et al., 2015). In addition, single space instruments are of limited lifetime, while long-term studies on ozone require combining measurements from different instruments, which could introduce uncertainty due to the differences and drifts among datasets (Rahpoe et al., 2015)…"*

8. Lines 68-72
A new area of global modelling involves the production of chemical reanalyses, which assimilate satellite data, to improve the quantification of tropospheric ozone, e.g. Miyazaki et al., 2020a,b; Colombi et al., 2021.
Response: Thanks for the suggestion. We have added the assimilation of satellite data using chemical models in Line 97-101:
*"Some advanced models can improve global tropospheric ozone in 3 dimensions by assimilating satellite data to enhance the modeling accuracy (Miyazaki et al., 2020a; Colombi et al., 2021). However, in addition to the aforementioned sources of uncertainties, such assimilations still rely on the sufficiency and spatial-temporal continuity of the satellite data (Huijnen et al., 2020; Miyazaki et al., 2020b)."*

9. Line 130
The Data and Methods section needs to state how the tropopause is defined, as the product is provided in terms of both the troposphere and stratosphere. If a forward or backward trajectory

begins in the troposphere and the final location of the trajectory particle, after 4 days, is above the tropopause, is this ozone observation categorized as being in the troposphere, or stratosphere?

Response: Thanks for the questions. Ozonesonde profiles are labelled according to their origin in the troposphere or stratosphere (defined by the WMO definition from the measured ozonesonde temperature profile). This allows us to generate forward and backward trajectories by only tropospheric or only stratospheric ozonesonde data to provide "troposphere-only" and "stratosphere-only" TOST fields, to be used with models, which generally calculate a tropopause. For the combined "trop_strat" product, no tropopause information is needed.

We described how we calculated the "troposphere-only" and "stratosphere-only" data and the definition of tropopause in Method in Line 232-238:

*"In addition, the ozonesonde data at the 26 levels are labelled according to their origin in the troposphere or stratosphere to generate forward and backward trajectories by using only tropospheric ozonesonde data or only stratospheric ozonesonde data. The tropopause is determined following the WMO definition from the measured ozonesonde temperature profile."*

10. Line 249-251

How were the IAGOS data averaged temporally? Into monthly means? What is the horizontal resolution? 5x5 degrees? How many airports were used? Did you use just the vertical profiles, or also the cruise level data? Do you have a data availability threshold? For example, do you require at least 4 profiles in a month to produce a monthly mean?

Response: Thanks for the questions. IAGOS data are averaged into monthly means with 1 km vertical resolution from sea level into horizontal bins of 5*5 degrees. In total, we used IAGOS data from 310 airports along the entire flight routines. We have not set data availability threshold when comparing the IAGOS data with TOST, i.e., the comparison is made for the grids having one or more data per month for both IAGOS and TOST.

The average of IAGOs data is rephrased with more details in Line 325-329:

*"Here, the IAGOS ozone profiles were processed into 1 km layers from sea level and averaged into bins of 5º latitude and 5º longitude for each month. In total, all IAGOS ozone data from 310 airports were used for the comparison (Table S2). Then, the processed IAGOS ozone profiles were matched with the TOST ozone for the corresponding grids to examine the performance of TOST in the troposphere."*

11. Line 540

The Guidance note on best statistical practices for TOAR analyses (described above) asks for all trends to be reported with 95% confidence intervals and p-values, and in units of ppbv decade-1. In the submitted manuscript trends are only reported for the stratosphere and in units of ppbv year-1. If ppbv year-1 is the standard unit for reporting ozone trends in the stratosphere, then please retain this unit, otherwise please follow the TOAR guideline.

Response: Thanks for the guidance. We have revised the trend by reporting the decadal trend with 95% confidence intervals in Line 633-634:

*"There are non-significant trends in the ozone concentrations at 21-22 km (by 0.5±0.6 %/decade) and 24-25 km (by -0.2±0.9 %/decade) from 1998 to 2021"*

12. Figure S3

Why compare IAGOS and TOST over the range 50-150 ppb which includes tropospheric and stratospheric samples? If a monthly mean value for IAGOS observations is 100 ppb then it is

very likely composed of both tropospheric samples (less than 100 ppb) and stratospheric samples (greater than 100 ppb). According to Figure S3d an IAGOS monthly mean of 100 ppb can correspond to a TOST value anywhere from 50 ppbv (mostly tropospheric samples) to 150 ppbv (mostly stratospheric samples). Clearly these two data sets are not sampling the same air masses and this is not an apples-to-apples comparison, so I don't see the value in these correlation plots. Compared only below tropopause.

Response: Thanks for the question. The separation of <50 ppb and 50-150 ppb here is to see the comparisons of IAGOS and TOST data in the lower and upper troposphere. However, we agree that comparing the range of 50-150 ppb could result in comparing different air masses sampling. Therefore, we only keep Figure 7 (the comparison of <50 ppb ozone samples) to make sure both IAGOS and TOST record tropospheric ozone, which is also the purpose of this figure: to compare the tropospheric ozone in TOST to another broadly trusted tropospheric ozone data (IAGOS). In addition, only the comparison of 1994-2021 was kept in Figure 7 for conciseness.

13. Section 4.2
Every year, stratospheric ozone trends and variability are updated in the Global Climate chapter of the State of the Climate reports. The most recent edition (Dunn et al., 2023) provides an update through the end of 2022. In particular, Figure 2.64 compares several products and shows stratospheric ozone levels at 22 km for the latitude band 35N-60N, similar to your Figure 13. The SWOOSH product (Davis et al., 2016) is a combined satellite product, bias corrected against ozonesonde observations and provides global coverage. How does TOST compare to these other products, and does TOST provide any new information?

Response: Thanks for the questions and comments. Although stratospheric ozone trends and variability are updated every year, the ozone trends in the lower stratosphere still have large uncertainties and differences (Ball et al., 2020; Li et al., 2023). Therefore, focusing on the lower stratosphere, we used TOST data to compare the lower stratospheric ozone trend (at 21-22km and 24-25km) since 1998 with recent studies. Over Northern Hemisphere mid-latitudes, the time series of the updated TOST shows an overall insignificant change in lower stratospheric ozone after 1998 (Figure 12), which is different from the decreasing trend reported using satellite-based data (Ball et al., 2018, 2019; Szeląg et al., 2020; Li et al., 2023). Therefore, more in-depth studies of stratospheric ozone trends, especially in the lower stratosphere, are necessary. Accordingly, we renamed Section 4.2 as "Long-term trend in the lower stratospheric ozone" and added the result in this section in Line 609-614:

*"Following the implementation of the Montreal Protocol and its amendments, recent studies have found an increase in upper stratospheric ozone since the late 1990s (Chipperfield et al., 2017; Szelag et al., 2020; Dunn et al., 2023). However, the lower stratospheric ozone trend remains highly uncertain (Ball et al., 2020). Quantifying lower stratospheric ozone trends depends largely on the quality of the observational datasets (Li et al., 2023)."*
And in Line 636-645:
*"There is no significant trend in the ozone concentrations at 21-22 km (0.5±0.6 %/decade) and 24-25 km (by -0.2±0.9 %/decade) from 1998 to 2021, indicating little change of lower stratospheric ozone, despite the fact that 25 years have passed since peak stratospheric chlorine. Recent studies using merged satellite data suggest that the decrease in the lower stratospheric ozone is offsetting the increase in the upper stratosphere (Ball et al., 2018, 2019; Szelag et al., 2020; Li et al., 2023. However, in the Northern Hemisphere mid-latitudes, TOST indicates no significant trend in the lower stratospheric ozone after the late 1990s. The differences between*

*satellite-based data and TOST call for further in-depth studies on the stratospheric ozone trend, especially in the lower stratosphere."*

[Figure]

*Figure 12. TOST time series of the annual mean ozone mixing ratios anomaly (in %) averaged over 30°-70°N over 21-22 km altitude (a) and 24-25 km altitude (b). The black dots represent the annual mean ozone concentrations from the area-weighted average of the grid cells over 30-70°N with ozone data throughout 1970-2021. The red line is the 3-year running mean.*

**References**

Ball, W. T., Chiodo, G., Abalos, M., Alsing, J., and Stenke, A.: Inconsistencies between chemistry–climate models and observed lower stratospheric ozone trends since 1998, Atmospheric chemistry and physics, 20, 9737-9752, 2020.

Ball, W. T., Alsing, J., Staehelin, J., Davis, S. M., Froidevaux, L., and Peter, T.: Stratospheric ozone trends for 1985–2018: sensitivity to recent large variability, Atmospheric Chemistry and Physics, 19, 12731-12748, 2019.

Ball, W. T., Alsing, J., Mortlock, D. J., Staehelin, J., Haigh, J. D., Peter, T., Tummon, F., Stübi, R., Stenke, A., and Anderson, J.: Evidence for a continuous decline in lower stratospheric ozone offsetting ozone layer recovery, Atmospheric Chemistry and Physics, 18, 1379-1394, 2018.

Harmens, H., Hayes, F., Mills, G., Sharps, K., Osborne, S., and Pleijel, H.: Wheat yield responses to stomatal uptake of ozone: Peak vs rising background ozone conditions, Atmospheric environment, 173, 1-5, 2018.

Li, Y., Dhomse, S. S., Chipperfield, M. P., Feng, W., Bian, J., Xia, Y., and Guo, D.: Quantifying stratospheric ozone trends over 1984–2020: a comparison of ordinary and regularized multivariate regression models, Atmospheric Chemistry and Physics, 23, 13029-13047, 2023.

Szeląg, M. E., Sofieva, V. F., Degenstein, D., Roth, C., Davis, S., and Froidevaux, L.: Seasonal stratospheric ozone trends over 2000–2018 derived from several merged data sets, Atmospheric Chemistry and Physics, 20, 7035-7047, 2020.

Vicedo-Cabrera, A. M., Sera, F., Liu, C., Armstrong, B., Milojevic, A., Guo, Y., Tong, S., Lavigne, E., Kyselý, J., and Urban, A.: Short term association between ozone and mortality: global two stage time series study in 406 locations in 20 countries, bmj, 368, 2020.